# Coronary blood vessels from distinct origins converge to equivalent states during mouse and human development

Ragini Phansalkar[1,2], Josephine Krieger[2], Mingming Zhao[3,4], Sai Saroja Kolluru[5,6], Robert C Jones[5], Stephen R Quake[5,6], Irving Weissman[7], Daniel Bernstein[3,4], Virginia D Winn[8], Gaetano D'Amato[2]*†, Kristy Red-Horse[2,4,7]*†

[1]Department of Genetics, Stanford University School of Medicine, Stanford, United States; [2]Department of Biology, Stanford University, Stanford, United States; [3]Division of Pediatric Cardiology, Department of Pediatrics, Stanford University School of Medicine, Stanford, United States; [4]Stanford Cardiovascular Institute, Stanford University School of Medicine, Stanford, United States; [5]Department of Bioengineering and Department of Applied Physics, Stanford University, Stanford, United States; [6]Chan Zuckerberg Biohub, Stanford, United States; [7]Institute for Stem Cell Biology and Regenerative Medicine, Stanford University School of Medicine, Stanford, United States; [8]Department of Obstetrics and Gynecology, Stanford University School of Medicine, Stanford, United States

*For correspondence:
damatog@stanford.edu (GD'A);
kredhors@stanford.edu (KR-H)

†These authors contributed equally to this work

Competing interest: The authors declare that no competing interests exist.

**Abstract** Most cell fate trajectories during development follow a diverging, tree-like branching pattern, but the opposite can occur when distinct progenitors contribute to the same cell type. During this convergent differentiation, it is unknown if cells 'remember' their origins transcriptionally or whether this influences cell behavior. Most coronary blood vessels of the heart develop from two different progenitor sources—the endocardium (Endo) and sinus venosus (SV)—but whether transcriptional or functional differences related to origin are retained is unknown. We addressed this by combining lineage tracing with single-cell RNA sequencing (scRNAseq) in embryonic and adult mouse hearts. Shortly after coronary development begins, capillary endothelial cells (ECs) transcriptionally segregated into two states that retained progenitor-specific gene expression. Later in development, when the coronary vasculature is well established but still remodeling, capillary ECs again segregated into two populations, but transcriptional differences were primarily related to tissue localization rather than lineage. Specifically, ECs in the heart septum expressed genes indicative of increased local hypoxia and decreased blood flow. Adult capillary ECs were more homogeneous with respect to both lineage and location. In agreement, SV- and Endo-derived ECs in adult hearts displayed similar responses to injury. Finally, scRNAseq of developing human coronary vessels indicated that the human heart followed similar principles. Thus, over the course of development, transcriptional heterogeneity in coronary ECs is first influenced by lineage, then by location, until heterogeneity declines in the homeostatic adult heart. These results highlight the plasticity of ECs during development, and the validity of the mouse as a model for human coronary development.

## Editor's evaluation

The authors largely improved the paper with additional analyses and extensive revisions. By employing scRNA-seq analyses, they elegantly have dissected the endothelial cell (EC) heterogeneity of cardiac blood vessels across development in mouse and human. They convincingly demonstrate that the EC heterogeneities of cardiac blood vessels are sequentially governed by the

progenitor sources and environmental cues during initial and late development. Moreover, they show that these ECs become homogeneous in adult. They also claim that human fetal hearts take on generally a similar path for establishment of cardiac blood vessels. Overall, this study is novel and intriguing.

## Introduction

During embryonic development, progenitor tissue sources produce new cell types through shifts in epigenetic and transcriptional states. Much research addresses how new cell types form, yet there is less focus on how the transcriptional or chromatin states of progenitor cells relate to gene expression in their descendants, that is, what do mature cells 'remember' about their history? This question is particularly intriguing in cases where multiple progenitor sources contribute to the same cell type, since different origins could result in different behaviors or responses to injury and disease. Such lineage merging is referred to as 'convergent differentiation' and occurs in hematopoietic populations (*Sathe et al., 2013*; *Weinreb et al., 2020*), oligodendrocytes (*Marques et al., 2018*), olfactory projection neurons (*Li et al., 2017*), coronary blood vessels of the heart (*Sharma et al., 2017*), and others (*Wei et al., 2015*; *Gerber et al., 2018*; *Konstantinides et al., 2018*). Recent single-cell RNA sequencing (scRNAseq) analyses have suggested that the resulting cell types can converge transcriptionally (*Li et al., 2017*), but in some cases maintain molecular signatures of their progenitors (*Dick et al., 2019*; *Weinreb et al., 2020*). However, there is no information on how convergent differentiation influences coronary blood vessels of the heart or how this might affect cardiac injury responses.

In this study, we investigated gene expression patterns in two lineage trajectories that form the coronary vasculature in mice and compared these with data from human fetal hearts. The major progenitor sources for coronary endothelial cells (ECs) in mice are the sinus venosus (SV), the venous inflow tract of the developing heart, and the endocardium (Endo), the inner lining of the heart ventricles (*Red-Horse et al., 2010*; *Wu et al., 2012*; *Chen et al., 2014*; *Tian et al., 2014*; *Zhang et al., 2016*; *Sharma et al., 2017*; *Su et al., 2018*). ECs begin to migrate from both these sources at embryonic day 11.5. They form an immature capillary plexus (*Zeini et al., 2009*; *Red-Horse et al., 2010*) by populating the heart with vessels from the outside-in (SV) or the inside-out (Endo). These two sources eventually localize to largely complimentary regions in adults: the SV contributes vessels to the outer myocardial wall and the Endo contributes vessels to the inner myocardial wall and the septum (*Red-Horse et al., 2010*; *Wu et al., 2012*; *Chen et al., 2014*; *He et al., 2014*; *Tian et al., 2014*; *Zhang et al., 2016*; *Sharma et al., 2017*). Initial angiogenesis from the SV or Endo is guided by different signaling factors (*Wu et al., 2012*; *Arita et al., 2014*; *Chen et al., 2014*; *Su et al., 2018*; *Payne et al., 2019*). However, in circumstances where SV angiogenesis is stunted, Endo-derived vasculature can expand into the outer wall to compensate for the vessel loss (*Sharma et al., 2017*).

In contrast to the well-characterized spatial differences between Endo and SV angiogenesis, there have been no comparisons of transcriptional states between Endo- and SV-derived coronary vessels in development or in adulthood. In addition, there is little information on whether coronary ECs in humans are also derived from these sources, or whether the transcriptional and functional states that human coronary ECs pass through during development match those in the mouse. Understanding any lineage-specific or species-specific traits would have important implications for approaches that reactivate developmental pathways to increase angiogenesis in injured or diseased human hearts (*Smart, 2017*; *Payne et al., 2019*).

Here, we used scRNAseq of lineage-traced ECs from mouse hearts at various stages of development to show that while SV- and Endo-derived capillary cells initially retained some source-specific gene expression patterns, these differences were only present at an early stage of development. Later, these lineages mixed into two capillary subtypes, which were correlated with different locations in the heart. By adult stages, SV- and Endo-derived capillary cells had converged into similar gene expression patterns, and differing lineage did not result in differential proliferation in response to ischemia/reperfusion (IR)-induced injury. Finally, scRNAseq on human fetal hearts indicated that human development closely matched that in mice, and provided additional insights into human coronary artery development. Based on our results, we propose a model in which the transcriptional state of non-proliferative coronary ECs is initially influenced by their lineage, then by regional differences in environmental factors, until both of these signatures fade in adulthood. These findings highlight the

importance of environmental factors in influencing EC behavior and validate mice as a representative model for human coronary development.

## Results

### ScRNAseq in lineage-labeled coronary ECs

To compare Endo- and SV-derived ECs during development and in adult hearts, we combined scRNAseq with lineage-specific fluorescent labeling (*Figure 1A and B*). A tamoxifen-inducible *Bmx^CreER* (*Ehling et al., 2013*) mouse was crossed with the *Rosa^tdTomato* Cre reporter, which specifically labels a high percentage of the Endo (94.44% of Endo cells labeled at e12.5), but does not mark the SV (3.61% of SV cells labeled at e12.5) (*D'Amato et al., 2021*). Labeling was induced before e11.5, when coronary development begins, so Endo-derived ECs expressed *tdTomato* while SV-derived ECs did not (Materials and methods). Cells from e12, e17.5, and adult hearts were sorted using fluorescence-activated cell sorting (FACS) and processed using the 10× Genomics platform (*Figure 1A–B* and *Figure 1—figure supplement 1A-E*). These strategies captured the expected EC subtypes at each stage (including coronary, valve, Endo, and SV) (*Figure 1—figure supplement 2A-D*) and contained a large number of cells that passed standard quality controls (Materials and methods). ScRNAseq analyses are most accurate when the specific cell populations of interest are extracted and re-analyzed without the influence of other cell types in the dataset, including cycling cells (*Luecken and Theis, 2019*). Thus, we isolated non-cycling coronary ECs (for e12—*Pecam1+, Cldn5+, Npr3-, Top2a-, Mki67, Tbx20-, Cldn11-, Bmp4-, Vwf-*; for e17.5—*Pecam1+, Npr3-, Tbx20-, Pdgfra-, Top2a-, Bmp4-, Mki67-*) (*Figure 1—figure supplement 2E and F*) and performed direct comparisons of cell states between Endo- and SV-enriched coronary ECs. The remaining cells in the dataset will be reported by D'Amato et al., which addresses experimental questions outside the scope of this study.

We first used unbiased clustering to identify coronary EC subtypes within the e12, e17.5, and adult datasets. Clustering resolution was determined individually for each dataset as the highest resolution at which every cluster expressed at least one unique marker gene. E12 coronary ECs separated into three clusters—capillary plexus 1 (Cap1), capillary plexus 2 (Cap2), and pre-artery (*Figure 1C*). Markers used to identify these populations matched previous reports and are shown in *Figure 1—figure supplement 3A*. The absence of venous ECs and the higher numbers of SV-enriched cells present at this stage are also consistent with previous studies (*Red-Horse et al., 2010*; *Su et al., 2018*). Separating plots by sample revealed that Cap2 was exclusively from the SV-enriched sample (*Figure 1D and E*). Consistent with this was its increased expression of *Aplnr* (*Apj*) (*Figure 1—figure supplement 3A*), which we previously demonstrated to be enriched in SV-derived vessels (*Sharma et al., 2017*). All but one of the Endo-enriched capillary cells were in Cap1, but Cap1 also contained cells from the SV-enriched sample (*Figure 1D and E*). Although recombination rates in the Endo were very high (Materials and methods), we cannot exclude the possibility that a small number of Endo-derived ECs are *tdTomato*-negative due to some un-recombined Endo ECs. These data show that shortly after coronary development is initiated, lineage is correlated with transcriptionally distinct capillary populations within the immature capillary plexus.

To test whether this phenomenon persists into late development, we similarly analyzed coronary ECs at e17.5. A larger number of coronary ECs were captured due to the increase in cardiac vasculature by this stage. EC clusters in this sample included two artery (Art1 and Art2), one vein, and two capillary (Cap1 and Cap2) clusters (*Figure 1F* and *Figure 1—figure supplement 3B*). If Cap1 and Cap2 continued to reflect different lineages, we would expect at least one cluster to contain only Endo- or SV-enriched ECs. There was skewed contribution with a higher percentage of Endo-enriched cells in Cap1 and a higher percentage of SV-enriched cells in Cap2 (*Figure 1G and H*), but no cluster was lineage exclusive, suggesting that additional factors were driving transcriptional heterogeneity. Veins were much more represented in the SV-enriched sample (*Figure 1H*), which is expected since most veins reside on the surface of the heart and SV angiogenesis progresses outside-in while Endo angiogenesis is in the opposite direction. We also considered the e17.5 dataset with cycling cells included, but with cell cycle effects regressed out, in order to evaluate whether there are differences in proliferation between Endo- and SV-enriched cells, and to rule this out as a cause for the difference in cluster distribution between the two lineages (*Figure 1—figure supplement 4A and B*). This analysis showed that cycling capillary cells also segregate into the Cap1 and Cap2 clusters (*Figure 1—figure*

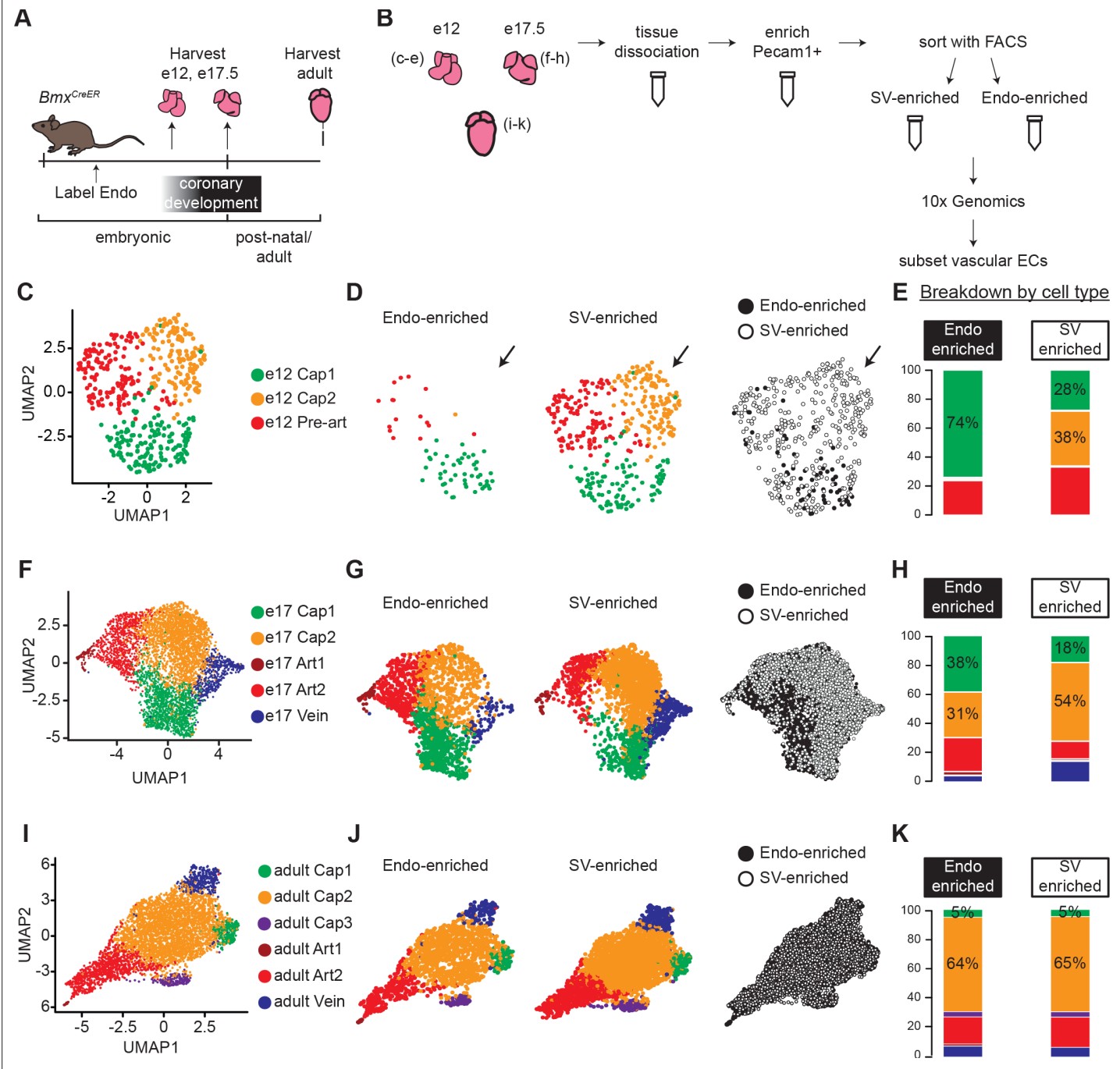

**Figure 1.** Single-cell RNA sequencing (ScRNAseq) of lineage-traced coronary endothelial cells (ECs) at three stages reveals capillary heterogeneity during embryonic development. (**A and B**) Overview of lineage tracing and scRNAseq approach in embryonic and adult mice. (**C–K**) Unbiased clustering of embryonic coronary ECs at the indicated time points and the contribution of endocardium (Endo)-enriched (*Bmx^CreER* lineage-labeled) and sinus venosus (SV)-enriched (*Bmx^CreER* lineage negative) cells to each cluster. Uniform Manifold Approximation and Projections (UMAPs) are shown for combined data (**C**, **F**, and **I**) and separated by lineage (**D**, **G**, and **J**) and percentages enumerated (**E**, **H**, and **K**).

The online version of this article includes the following figure supplement(s) for figure 1:

**Figure supplement 1.** Localization and expression of recombinant markers.

**Figure supplement 2.** Selection of coronary vascular endothelial cells (ECs) from e12 and e17.5 datasets.

**Figure supplement 3.** Coronary endothelial cell (EC) subtype markers.

**Figure supplement 4.** Cell cycle regression in e17.5 coronary endothelial cells (ECs).

**Figure supplement 5.** Expression of marker genes adult mouse coronary endothelial cell (EC) dataset.

supplement 4C), and that their distribution into these clusters is biased by lineage, similar to non-cycling cells (*Figure 1—figure supplement 4D*). Additionally, there is no difference in proliferation between Cap1 and Cap2 (*Figure 1—figure supplement 4E*).

We next performed the same analyses on adult coronary ECs. Similar to e17.5, clustering revealed one vein and two artery clusters. It additionally revealed three capillary clusters, Cap1, Cap2 and Cap3 (*Figure 1I*), which were distinguishable by expression of unique gene markers (*Figure 1—figure supplement 3C* and *Figure 1—figure supplement 5A*). While Cap1 contained the majority of capillary cells, Cap2 was distinguished by expression of pro-angiogenic genes including *Apln* and *Adm*, and Cap3 was distinguished by expression of interferon-induced genes such as *Ifit3* (*Figure 1—figure supplement 5A*). Both of these clusters have been reported previously in adult hearts (*Kalucka et al., 2020*). However, unlike the clusters at earlier stages, there was a similar distribution of cells into each of these clusters in both the Endo- and SV-enriched samples (*Figure 1j–k*), and there were no appreciable gene expression differences between the samples. This observation is consistent with another lineage-specific adult scRNAseq dataset we produced with Smart-seq2 (unpublished results). Thus, we concluded that there is no lineage-based heterogeneity in adult coronary ECs.

## Coronary heterogeneity is first related to lineage and then to location

We next investigated the genes driving coronary ECs into two capillary cell states at the different stages of development. One hypothesis was that cells retained gene expression patterns from their progenitors. To test this, a list of genes defining the progenitor states (the Endo and SV) was compiled by directly comparing gene expression in the Endo and the SV at e12 (*Figure 1—figure supplement 2A*) and using differentially expressed genes (DEGs) passing significance thresholds described in the Materials and methods (*Table 1*). The expression of these genes was then assessed in capillary clusters. At e12, Cap1 cells expressed higher levels of Endo-specific genes, while Cap2 cells expressed higher levels of SV-specific genes (*Figure 2A* and *Figure 2—figure supplement 1A-C*). Indeed, 40% of Cap1 genes and 3% of Cap2 genes overlapped with the Endo, while 47% of Cap2 genes and 1% of Cap1 genes overlapped with the SV (*Figure 2B*). We concluded that the transcriptional identities of Cap1 and Cap2 cells derive at least in part by gene expression patterns retained from the SV or Endo.

**Table 1.** Differentially expressed genes between endocardium (Endo) and sinus venosus (SV) (from heatmap in *Figure 2A*).

| Endo-specific genes | SV-specific genes |
| --- | --- |
| Cdkn1c | Cldn11 |
| Tmem108 | Aplnr |
| Nrk | Agr2 |
| Fabp5 | Bmp4 |
| Cdh11 | Hoxa5 |
| Cd81 | Gm13889 |
| Irx5 | Zfp503 |
| H19 | Hotairm1 |
| Hand2 | Rassf9 |
| Adgrg6 | Mmrn1 |
| Maged2 | Pcdh17 |
| Ccnd2 | Tox |
| Plvap | Slco3a1 |
| Igf2 | Tspan13 |
| Dok | Nr2f2 |
| Col13a1 | Fst |
| Tm4sf1 | Cd36 |
| Igf2r | Cldn5 |
| Gm1673 | Aqp1 |
| Blvrb | Hoxb4 |
| Gsta4 | Khdrbs3 |
| Rap2a | Kitl |
| Gatm | Wnt16 |
| Olfml3 | Limch1 |
| Sdpr | Ahr |
| Ece1 | Edn1 |
| Plagl1 | Lamp5 |
| Prr15 | Cav1 |
| Igfbp4 | Ddah1 |
| Ccnd3 | Tbx5 |

The online version of this article includes the following source data for table 1:

**Source data 1.** List of all differentially expressed genes between e12 endocardium (Endo) and sinus venosus (SV) sorted by Wilcoxon rank sum test.

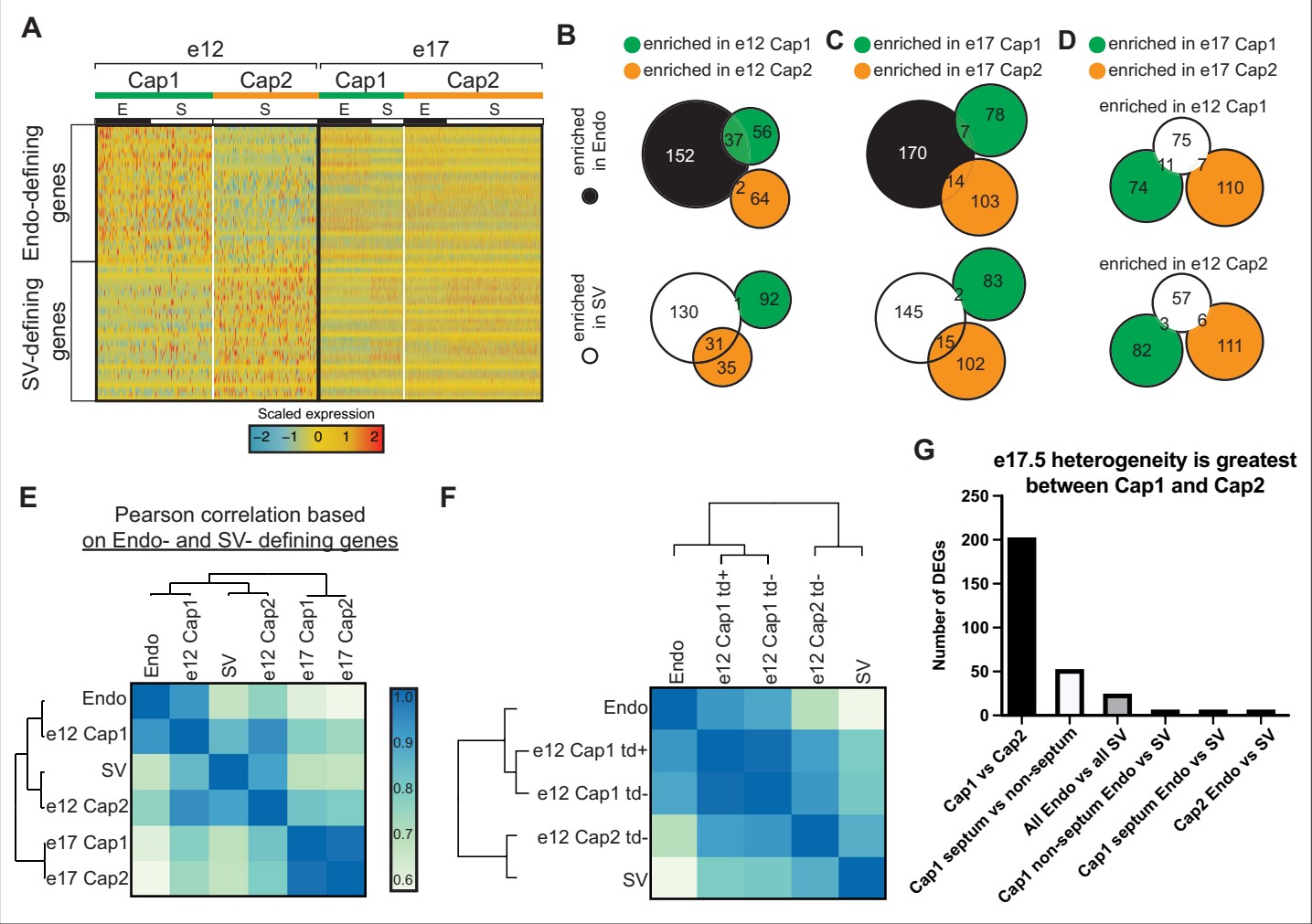

**Figure 2.** Expression of endocardium (Endo) and sinus venosus (SV) genes in coronary endothelial cells (ECs). (**A**) Heatmap showing expression of the top 30 (by p-value) Endo-defining genes (enriched in the Endo compared to the SV) and the top 30 (by p-value) SV-defining genes (enriched in the SV compared to the Endo) in e12 and e17.5 capillary clusters (E = coronary cells from the Endo-enriched sample, S = coronary cells from the SV-enriched sample). (**B and C**) Venn diagrams showing overlap of Endo- and SV-defining genes with Cap1-enriched genes (enriched in Cap1 compared to Cap2) and e12 Cap2-enriched genes (enriched in Cap2 compared to Cap1) at e12 (**B**) and e17.5 (**C**). (**D**) Venn diagram showing overlap of e12 Cap1- and Cap2-enriched genes with e17.5 Cap1 and Cap2 genes. (**E and F**) Heatmaps of Pearson correlations based on expression of Endo- and SV-defining genes in the Endo, the SV, and capillary clusters from e12 and e17.5 in total (**E**) and separated by *Bmx*^CreER lineage as indicated by *tdTomato* (*td*) expression (**F**). (**G**) Bar plot showing number of differentially expressed genes (DEGs) between different subgroups of capillary cells.

The online version of this article includes the following figure supplement(s) for figure 2:

**Figure supplement 1.** Expression of selected endocardium (Endo)- and sinus venosus (SV)-defining genes.

**Figure supplement 2.** Expression of e12 Cap1- and Cap2-specific genes in a dataset of e12.5 sinus venosus (SV)-derived endothelial cells (ECs).

This pattern was not observed at e17.5. There was no clear pattern between e17.5 Cap1 and Cap2 in the expression of SV- and Endo-specific genes (*Figure 2A* and *Figure 2—figure supplement 1D*), and there was little to no overlap between Cap1 and Cap2 differential genes and SV or Endo genes (*Figure 2C*). Although it may appear on the heatmap in *Figure 2A* that there is a lineage-based distinction in Cap1 for a small subset of the genes at e17.5, this is not the case. As we will demonstrate later in *Figure 3—figure supplement 2*, the data indicated that these results were primarily due to Cap1 containing septum ECs, which are mostly derived from the Endo, and the septum imparting a location-specific effect on transcription. Furthermore, the minimal overlap between the e12 and e17.5 Cap1- and Cap2-defining genes is consistent with e17.5 coronary ECs not retaining the progenitor-type genes enriched in e12 clusters (*Figure 2D*).

Calculating Pearson correlations using Endo and SV genes revealed that e12 coronary ECs were similar to their progenitor sources while e17.5 coronary ECs were much less so (*Figure 2E*). We also calculated Pearson correlations as a function of lineage at e12. As expected, *tdTomato*-positive cells were highly similar to the Endo while *tdTomato*-negative cells in Cap2 were more similar to the SV than to the Endo (*Figure 2F*). Interestingly, *tdTomato*-negative cells in Cap1 were more similar to the Endo than to the SV (*Figure 2F*). This could result from a number of reasons that we cannot currently distinguish including: (1) SV-derived cells migrating close to the Endo take on Endo-type gene expression or (2) there is a rare Endo population that does not express *Bmx*. Point 2 is supported over point 1 due to the observation that Cap2-like cells are present in a previous scRNAseq dataset of SV-derived ECs (*Su et al., 2018*), while Cap1 cells are not (*Figure 2—figure supplement 2A-E*). In total, these data indicate that e17.5 capillary heterogeneity is not a remnant of the heterogeneity present at e12, and lineage-related differences are not apparent in adults.

Further analyses provided additional evidence that the differences between e17.5 Cap1 and Cap2 are not primarily due to lineage. With the prediction that significant lineage-related heterogeneity would be accompanied by substantial differences in gene expression, we compared the DEGs

**Table 2.** Differentially expressed genes between all endocardium (Endo)-enriched and all sinus venosus (SV)-enriched capillaries.
Bolded genes are also differentially expressed between Cap1 and Cap2.

| | Raw p-value | Average log-fold change | % Endo cells expressing | % SV cells expressing | Adjusted p-value | Higher fold change in Cap1 versus Cap2 comparison? |
|---|---|---|---|---|---|---|
| *tdTomato* | 1.91E-238 | 0.39378511 | 0.452 | 0 | 5.35E-234 | |
| *Gt(ROSA)26Sor* | 2.43E-153 | –0.634801942 | 0.568 | 0.837 | 6.81E-149 | |
| **Anxa1** | 4.24E-109 | 0.553933954 | 0.804 | 0.53 | 1.19E-104 | |
| *Igf2* | 6.74E-93 | –0.357191579 | 0.99 | 0.999 | 1.89E-88 | |
| **Timp4** | 1.28E-88 | –0.632710228 | 0.447 | 0.766 | 3.58E-84 | Yes |
| **Fmo1** | 8.34E-70 | –0.351754806 | 0.231 | 0.511 | 2.34E-65 | Yes |
| **Txnip** | 2.09E-67 | –0.37131915 | 0.851 | 0.933 | 5.85E-63 | Yes |
| **Aplnr** | 5.42E-67 | –0.40125981 | 0.677 | 0.863 | 1.52E-62 | Yes |
| **Car4** | 1.68E-65 | –0.595662824 | 0.347 | 0.62 | 4.70E-61 | Yes |
| **Aqp7** | 3.33E-62 | –0.368823615 | 0.345 | 0.63 | 9.31E-58 | Yes |
| *Apoe* | 1.19E-60 | 0.302862096 | 0.307 | 0.09 | 3.34E-56 | |
| **Cd36** | 8.58E-59 | –0.308565577 | 0.97 | 0.997 | 2.40E-54 | Yes |
| *Igfbp5* | 1.84E-57 | 0.4602195 | 0.258 | 0.064 | 5.15E-53 | |
| **Fabp5** | 6.53E-54 | 0.30861622 | 0.999 | 1 | 1.83E-49 | Yes |
| **1810011O10Rik** | 1.08E-52 | –0.340249285 | 0.907 | 0.963 | 3.04E-48 | Yes |
| **Aqp1** | 9.79E-47 | –0.471913791 | 0.54 | 0.729 | 2.74E-42 | Yes |
| **Gap43** | 7.23E-38 | 0.34418651 | 0.523 | 0.331 | 2.02E-33 | Yes |
| **Sat1** | 1.99E-37 | 0.394143816 | 0.753 | 0.63 | 5.59E-33 | Yes |
| **Cd63** | 3.81E-36 | 0.33008301 | 0.758 | 0.611 | 1.07E-31 | Yes |
| **Igfbp3** | 3.45E-34 | 0.640208849 | 0.634 | 0.468 | 9.66E-30 | Yes |
| *Tm4sf1* | 1.38E-32 | 0.302207793 | 0.982 | 0.94 | 3.87E-28 | |
| *Maged2* | 4.03E-32 | 0.323885527 | 0.845 | 0.757 | 1.13E-27 | |
| **Ly6c1** | 1.32E-31 | –0.301696479 | 0.845 | 0.93 | 3.70E-27 | Yes |
| **Rbp1** | 1.26E-15 | 0.432393695 | 0.757 | 0.69 | 3.52E-11 | Yes |

between e17.5 Cap1 and Cap2 to the DEGs between Endo- and SV-enriched capillary cells. There were 202 DEGs between e17.5 Cap1 and Cap2, but only 24 DEGs between all Endo-enriched and SV-enriched capillaries (*Figure 2H*). Inspecting DEG identities provided further support that lineage is not retained. Eighteen of the DEGs between all Endo- and SV-enriched cells were also DEGs between Cap1 and Cap2 (*Table 2*). If the differential patterns of these genes were due to a lineage effect, we would expect a greater log-fold change in the all Endo- versus SV-enriched comparison than in the Cap1 versus Cap2 comparison. However, 16 of the 18 genes have a greater log-fold change in the Cap1 versus Cap2 comparison (*Table 2*). These data indicate that differential expression between the Endo- and SV-enriched capillaries mostly stems from the differential contribution of the Endo and SV lineages to Cap1 and Cap2.

Since there was not strong evidence that lineage was a significant factor, we next considered whether differential localization in the heart might underlie e17.5 heterogeneity. During development, different regions of the heart show varying levels of hypoxia and signaling factors, for example, at some stages, the septum is more hypoxic and expresses higher levels of *Vegfa* (*Miquerol et al., 2000*; *Sharma et al., 2017*). Localization-driven heterogeneity would also explain the bias in cluster distribution between the Endo and SV lineages (*Figure 1H*) because they contribute ECs to complementary regions of the heart. This is most dramatic in the septum where almost all ECs are derived from the Endo (*Wu et al., 2012*; *Chen et al., 2014*; *Zhang et al., 2016*). The top DEGs between e17.5 Cap1 and Cap2 included hypoxia-induced genes (*Mif, Adm, Igfbp3, Kcne3*) (*Tazuke et al., 1998*; *Lee et al., 1999*; *Keleg et al., 2007*; *Simons et al., 2011*; *Heng et al., 2019*) and tip-cell markers (*Apln, Plaur, Lamb1, Dll4*) (*Hellström et al., 2007*; *del Toro et al., 2010*) in Cap1, and flow-induced genes (*Klf2, Klf4, Thbd, Lims*) (*Dekker et al., 2002*; *Hamik et al., 2007*; *Kumar et al., 2014*; *Wang and Zhang, 2020*) in Cap2 (*Figure 3A and B*, *Figure 3—figure supplement 1A*), suggesting pathways that could be consistent with localization-driven heterogeneity.

We next used Car4 to localize Cap2 within the tissue because it was specifically expressed in this cluster and there was an antibody available for immunostaining (*Figure 3A and C*). Erg-positive ECs expressing Car4 were located mainly in the right and left ventricle-free walls and dorsal side while ECs in the septum and ventral wall were mostly Car4-negative (*Figure 3D–F* and *Figure 3—figure supplement 1B*). This indicated that Cap1 localizes primarily to the septum and ventral wall and Cap2 to the remaining walls of the ventricle. The *Kcne3* pattern provided further support for this localization because it was specific to Cap1, and in situ hybridization revealed specific septal and ventral expression (*Figure 3G*). *Car4* analysis revealed a further segregation of Cap1 into septum and non-septum regions on the UMAP. Specifically, one side of the Cap1 cluster was almost completely devoid of *Car4* and comprised almost all Endo-enriched cells (*Figure 3C*), which are two features specific to the septum as shown in *Figure 3D–F* and *Figure 3—figure supplement 1B*. Additionally, these cells have lower expression of *Aplnr* compared to the non-septal cells of Cap1 (*Figure 1—figure supplement 3B*), which is also characteristic of the septum (unpublished observations). Although unbiased clustering did not distinguish this group of cells as a separate cluster, this could be because we used a combination of gene expression, protein staining, and lineage information (*Figure 3B–G*), which the clustering algorithms do not take into account. Interestingly, this proposed septum region is where gene expression indicates both decreased blood flow and local hypoxia (*Figure 3B*), the former of which would cause the latter. Thus, the coronary vasculature in the septum (and where the rest of Cap1 localizes) may not receive full blood flow at this late stage in development.

With the knowledge that half of Cap1 was almost completely represented by Endo-derived cells from the septum, we revisited the issue of lineage-based heterogeneity by manually searching for genes enriched in the proposed septum and analyzing whether they were correlated with lineage. A few of the Endo-specific genes from the heatmap in *Figure 2A* were expressed at a higher level in the septum while a few SV-specific genes were expressed at a lower level (*Figure 3—figure supplement 2A-D*). If this is due to retention of lineage information, we would expect to see the Endo-specific genes expressed in higher percentages of Endo-enriched cells compared to SV-enriched cells both inside and outside the septum. However, there was generally a decrease in the expression of these genes in both Endo- and SV-enriched cells outside of the septum, supporting the location, but not the lineage, hypothesis. Only two genes somewhat followed a pattern indicating lineage retention—*Gucy1b3* and *Hand2*— but these differences did not pass our pre-set significance thresholds

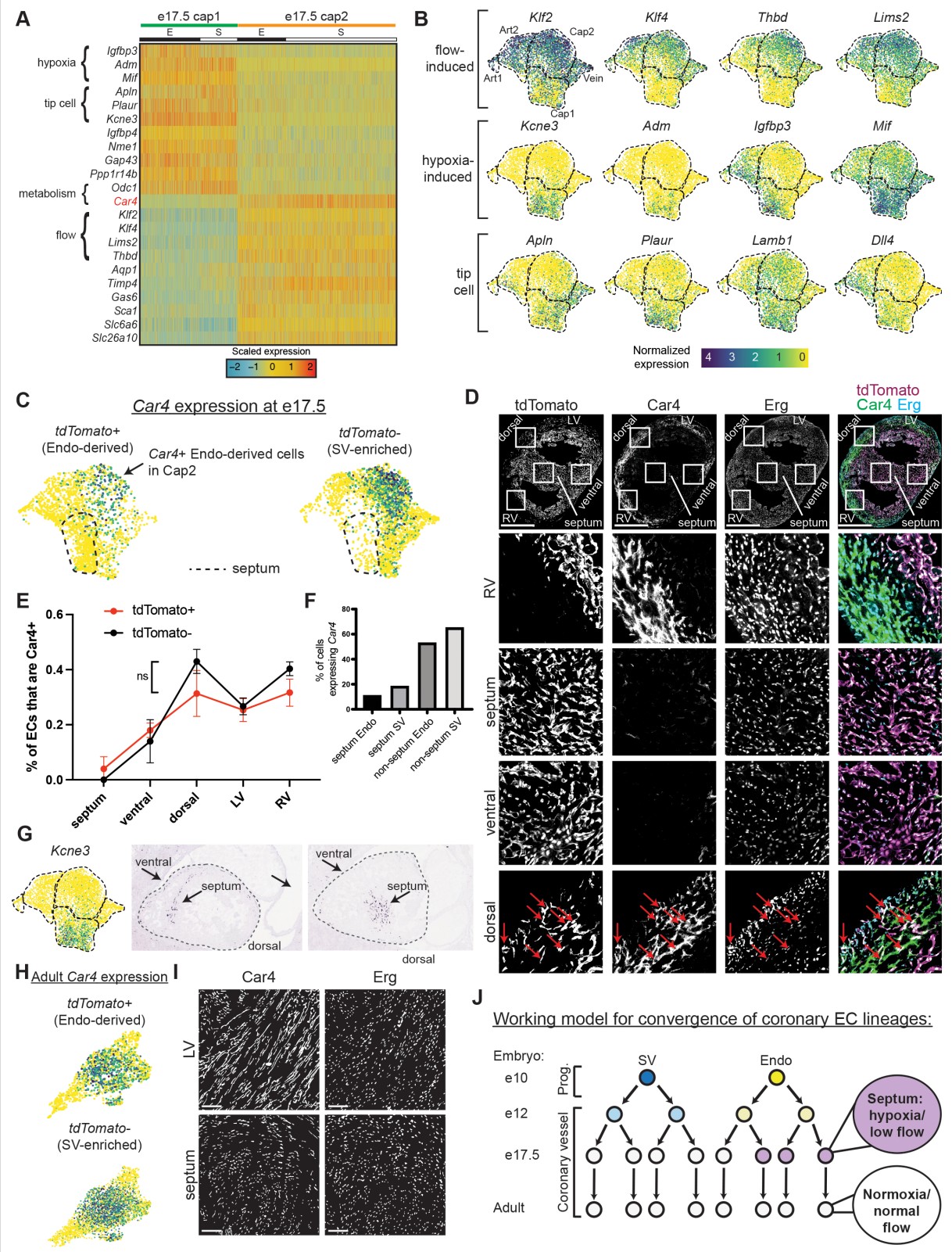

**Figure 3.** Gene expression and localization of e17.5 capillary clusters. (**A**) Heatmap showing expression of selected genes enriched in either Cap1 or Cap2 at e17.5 (E = coronary cells from endocardium [Endo]-enriched sample, S = coronary cells from sinus venosus [SV]-enriched sample). (**B**) Uniform Manifold Approximation and Projections (UMAPs) showing expression of selected flow-induced, hypoxia-induced, and tip-cell genes. Dashed lines outline indicated clusters. (**C**) *Car4* UMAPs separated by lineage. Dashed line shows area of UMAP enriched in Endo-enriched, *Car4*-negative cells

*Figure 3 continued on next page*

*Figure 3 continued*

predicted to be located in the septum. (**D**) Immunofluorescence of Car4 and Erg in a heart section from an e17.5 *Bmx^CreER^;Rosa^tdTomato^* embryo (scale bar = 500 μm). Red arrows indicate Car4-positive, tdTomato-positive Endo-derived ECs in the dorsal wall. (**E**) Plot showing percentage of tdTomato-positive and tdTomato-negative endothelial cells (ECs) in different locations which are also Car4-positive based on quantification of Car4 staining in Erg-positive cells from three e17.5 *Bmx^CreER^;Rosa^tdTomato^* embryos (error bars = range). (**F**) Bar plot based on e17.5 scRNAseq showing the percent of capillary cells in different categories (septum Endo-enriched, septum SV-enriched, non-septum Endo-enriched, non-septum SV-enriched) which express *Car4* at any level. (**G**) Images showing in situ hybridization for *Kcne3* in an e14.5 embryonic mouse heart, obtained and adapted from GenePaint (set ID EH3746). (**H**) UMAPs showing expression of *Car4* in adult coronary ECs, separated by lineage. (**I**) Immunofluorescence of Car4 and Erg in the left ventricle (LV) and septum of an adult wild-type (WT) heart (scale bar = 100 μm). (**J**) Working hypothesis for convergence of Endo- and SV-derived ECs into equivalent transcriptional states. Scale bar from (**B**) also applies to (**C**), (**G**), and (**H**).

The online version of this article includes the following source data and figure supplement(s) for figure 3:

**Source data 1.** Counts of Car4+ and Car4- endothelial cells in sections of *Bmx^CreER^* hearts at e17.5.

**Figure supplement 1.** Expression of flow-induced genes.

**Figure supplement 2.** Expression of endocardium (Endo)- and sinus venosus (SV)-defining genes at e17.5.

(Materials and methods) and were expressed in a low percentage of cells (<20% outside the septum) (*Figure 3—figure supplement 2B*).

We performed additional analysis comparing the number of DEGs between Endo- and SV-enriched cells within different capillary subgroups defined by transcriptional states (i.e. clustering) and in different locations, specifically, the proposed septal and non-septal cells of Cap1 and Cap2 (*Figure 2H*). If lineage was a major contributor to e17.5 heterogeneity, we would expect to see a substantial number of DEGs between Endo- and SV-enriched cells within a specific location. Instead, once the effect of location was removed by only comparing Endo- and SV-enriched cells either inside or outside of the septum, there were only six DEGs between the lineages (*Figure 2H*). Further supporting the impact of location in transcription, the second largest number of DEGs was between the septum and non-septum cells of Cap1 (*Figure 2H*). Thus, although we cannot exclude that a few genes retain lineage information in a small number of cells, the overwhelming evidence of this analysis supports that location has a greater effect on cell state at e17.5, with little to no contribution by lineage.

To probe whether the septal portion of Cap1 (see *Figure 3C*, dotted line) segregates based on cell-autonomous, lineage-specific features of Endo-derived ECs or regional environments, we took advantage of the fact that some Endo-lineage-labeled cells migrate into SV-biased territories during development (*Chen et al., 2014*). Cell-autonomous lineage differences would be supported if these cells remained Car4-negative in the ventricular walls. However, the opposite was true. Endo-lineage cells outside the septum were more likely to express Car4 than those in the septum (on average, 23% outside septum versus 4% inside septum) (*Figure 3F*, *Figure 3—figure supplement 1B*). Endo-derived ECs in SV-biased territories such as the dorsal side of the heart start to express Car4 (*Figure 3D*, arrows), and they can exist in the Cap2 state (*Figure 3C*, arrow). In addition, the rates of Car4 positivity in both Endo- and SV-enriched ECs correlate similarly with location in the heart (*Figure 3E*). Thus, the combination of lineage labeling, scRNAseq, and histology allowed us to ascertain that Endo-lineage ECs are biased toward a separate cell state based on their preferential location in hypoxic regions. However, due to the strong association between EC lineage and location, we cannot rule out that some minor degree of lineage-based heterogeneity exists at e17.5.

Since regional hypoxia would be incompatible with adult heart function, we predicted that the absence of strong heterogeneity in adult capillary ECs could be explained by a resolution of regional environmental differences after development. To test this, we examined *Car4* in the adult dataset, and found that it is broadly expressed in all adult capillary populations (*Figure 3H*). Additionally, immunostaining confirmed that septum ECs had become positive for Car4 in adults (*Figure 3I*, *Table 3*). Altogether, our data support the following model—that over the course of development, transcriptional

**Table 3.** Ratio of Erg+ Car4+/Erg+ cells in adult mouse hearts.

|  | LV | Septum |
| --- | --- | --- |
| heart1 | 0.874 | 0.853 |
| heart2 | 0.864 | 0.819 |

heterogeneity in coronary ECs is first influenced by lineage, then by location, until both lineage- and location-based heterogeneity disappear in the static adult heart (*Figure 3J*).

## Lineage does not change response to IR injury

Although Endo- and SV-derived capillary ECs were transcriptionally very similar in normal adult hearts, they could behave differently when challenged with hypoxia or other injury. To test this, we performed an IR injury by temporarily occluding the left anterior descending (LAD) coronary artery in adult *Bmx^CreER^;Rosa^tdTomato^* mouse hearts in which the Endo was labeled prior to coronary development (*Figure 4a* and *Figure 1—figure supplement 1b*). Proliferation in Endo- and SV-enriched ECs was quantified using EdU incorporation assays on day 5 post-injury. This time point was chosen based on prior data showing regrowth of the majority of coronary vessels 5 days after IR injury (*Merz et al., 2019*). Observing overall EdU incorporation revealed the site of injury, which was prominent in the mid-myocardial region between the inner and outer wall (*Figure 4B*). Proliferation was assessed in areas just below the ligation and apex of the heart, as this is the expected distribution of ischemia following LAD ligation (*Merz et al., 2019*). Regions of interest (ROIs) were chosen in myocardial areas containing a mix of Endo- and SV-derived ECs to ensure adequate representation from both lineages (*Figure 4C and D*). This analysis found no difference between the lineages (*Figure 4E*). We next performed EdU quantification in the inner and outer walls of the myocardium where most ECs derive from either the Endo or SV, respectively (*Wu et al., 2012*; *Tian et al., 2014*; *Zhang et al., 2016*; *Sharma et al., 2017*; *Figure 4B* and *Figure 1—figure supplement 1B*). In contrast to the lineage comparison, the outer wall showed a consistent, though non-significant, increase in EC proliferation over the inner wall, both within and adjacent to the injury site and regardless of lineage, indicating that this injury and proliferation assay can reveal differences (*Figure 4F*). These data support the notion that the local environment, rather than lineage, regulates capillary responses to injury in adult hearts, at least with respect to EC proliferation in the days after injury.

ECs exit the cell cycle during their differentiation into mature coronary arteries (*Fang et al., 2017*; *Su et al., 2018*). As a result, the proliferation of artery ECs is rare in the normal adult heart vasculature. Because IR injury induced the proliferation of capillary ECs, we investigated whether IR injury also induced the proliferation of artery ECs (*Figure 4G*). Approximately 69% of large arteries (n = 32) identified in the injured regions that we analyzed (as indicated in *Figure 4B*) contained at least one EdU-positive EC, and there was a significantly higher rate of EdU-positive ECs in the arteries of injured compared to uninjured hearts (*Figure 4H*). This observation could be due either to proliferating capillary cells which transitioned into artery cells but retained EdU or to artery ECs which began proliferating in response to the injury. Further studies will be necessary to determine whether one or both of these processes are occurring.

## Analogous features in mouse and human coronary ECs

We next sought to investigate whether comparing mouse and human scRNAseq datasets could provide insights into human development. ScRNAseq was performed using Smart-seq2 on PECAM1-positive ECs sorted from human fetal hearts at 11, 14, and 22 weeks of gestation. In addition, a capillary-specific marker, CD36, allowed for enrichment of PECAM1+ CD36- arterial ECs (*Figure 5A* and *Figure 5—figure supplement 1A and B*; *Cui et al., 2019*). After initial filtering, 2339 high-quality, high-coverage single EC transcriptomes were obtained, of which 713 were arterial. The data included 12 clusters of the expected cell types—artery, capillary, vein, cycling, Endo, valve EC—identified by known markers (*Figure 5B* and *Figure 5—figure supplement 1C*). As with the mouse data, the analysis was restricted to non-cycling arteries, capillaries, and veins in order to specifically compare cell states and trajectories in coronary vessels. Similar to mouse, the data contained one vein and two capillary clusters, but in contrast to mouse, there was an additional arterial cluster for a total of three (*Figure 5C*). There was approximately equal representation of the clusters in the PECAM1-positive fraction at each gestational age, consistent with the data being from later stages (mouse equivalent of e15.5–18.5)(*Krishnan et al., 2014*) when the developing heart is growing in size rather than producing new coronary cell subtypes (*Figure 5D* and *Figure 5—figure supplement 2A*). Consequently, we pooled data from the three time points for all further analyses. To begin comparing human and mouse, the *Seurat* Label Transfer workflow (*Stuart et al., 2019*) was utilized to reference map each human cell to its closest mouse cluster and vice versa (*Figure 5E*). This showed close concordance between the

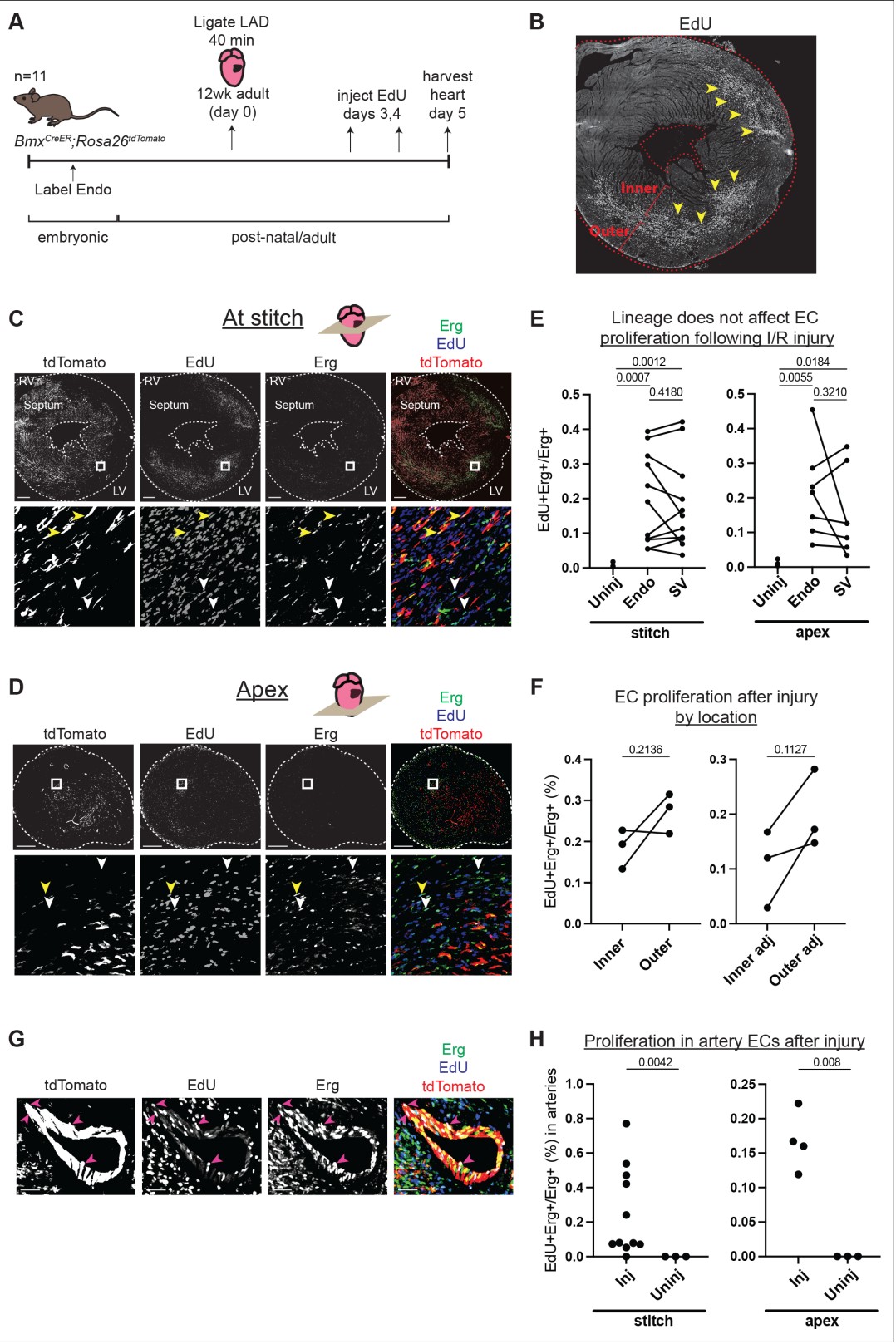

**Figure 4.** Comparison of injury responses of endocardium (Endo)- and sinus venosus (SV)-derived coronary endothelial cells (ECs). (**A**) Overview of lineage tracing and ischemia-reperfusion (I/R) injury approach in adult mice. (**B**) Example of how EdU localization highlights mid-myocardial injury region. Yellow arrowheads indicate the injury region with dense EdU staining. (**C and D**) Immunofluorescence of EdU and Erg in sections of the heart

*Figure 4 continued on next page*

*Figure 4 continued*

from (**B**) just below level of the stitch (**C**) and in the apex (**D**). Yellow arrowheads show proliferating Endo-derived ECs that are positive for tdTomato, Erg, and EdU; white arrowheads show tdTomato-negative, Erg-positive ECs from the SV that are EdU positive. (**E**) Quantification in multiple injured hearts of EdU-positive, Erg-positive ECs from the two lineages. (**F**) Quantification in multiple injured hearts of EdU-positive, Erg-positive ECs from the inner and outer wall, both in the focal area of the injury, as indicated in (**B**), and in areas adjacent to the injury. (**G**) Immunofluorescence of EdU and Erg in an artery of an injured heart. Pink arrowheads show proliferating ECs that are positive for tdTomato, Erg, and EdU. (**H**) Quantification in multiple injured hearts of EdU-positive, Erg-positive ECs in arteries in the focal area of the injury. In (**E**), (**F**), and (**H**), each dot represents one heart. Scale bar = 50 μm for (**G**). Scale bars = 500 μm for all other images.

The online version of this article includes the following source data for figure 4:

**Source data 1.** Counts of proliferating and non-proliferating endothelial cells in sections of adult *Bmx^CreER* hearts after injury.

two species (*Figure 5F and G*). With respect to the two capillary clusters that were extensively studied above, the majority of human Cap1 cells mapped to mouse Cap1, while the majority of human Cap2 cells mapped to mouse Cap2 (*Figure 5F*). When assigning mouse cells to human clusters, mouse Cap2 almost completely mapped to human Cap2, while mouse Cap1 mapped substantially to human Cap1, Cap2, and Art3 (*Figure 5G*). Analyzing specific gene expression revealed several enriched genes shared between corresponding mouse and human capillary clusters (including *KIT*, *ODC1*, *CD300LG*, *RAMP3*), and showed that human Cap1, like its mouse correlate, displayed patterns indicative of experiencing low blood flow conditions and potentially increased hypoxia, that is, lower *LIMS2, THBD, KLF2, KLF4* and higher *ADM, IGFBP3, KCNE3, LAMB1* (*Figure 5H* and *Figure 5—figure supplement 1D*). Notably, *CA4* (the human homolog of mouse *Car4*) was not differentially expressed between human Cap1 and Cap2.

Knowing that Cap1 and Cap2 segregate spatially in the mouse heart (*Figure 3D–F*), we examined whether human Cap1 and Cap2 also represent cells in different locations. We performed in situ hybridization for *TINAGL1*, a gene which is enriched in human Cap2 (*Figure 5I*). This revealed a statistically significant difference in the amount of *TINAGL1* RNA detection between the septum and the heart walls in both an 18-week and a 20-week gestational heart, with the septum having dramatically lower expression (*Figure 5J–K*). This difference was especially pronounced between the septum and the right ventricular free wall, similar to the Car4 pattern in mouse (*Figure 3E* and *Figure 3—figure supplement 1B*). In contrast to mouse Car4, there was no difference in *TINAGL1* detection between the ventral and dorsal walls (*Figure 5K*). Thus, in situ hybridization with *TINAGL1* supports a bias in the localization of human Cap1 to the septum and human Cap2 to the ventricular wall. Since lineage data and in situ immunofluorescence confirmed a subset of mouse Cap1 as containing the Endo-derived cells present in the septum (*Figure 3C*), the human expression data in *Figure 5H–K* suggested that human Cap1 may also represent an enrichment of septal cells with less blood flow and could also be biased toward the Endo lineage, although the latter cannot be confirmed with gene expression alone. When using the Label Transfer workflow to specifically map the putative mouse septal ECs to the human data, a higher percentage of human Cap1 than Cap2 cells matched to this septal EC group (35% of Cap1 cells versus 14% of Cap2 cells) (*Figure 5—figure supplement 1E*). We concluded that similar developmental environments and EC states exist between mouse and human capillaries, including those unique to the septum.

## Characterization of the capillary-to-artery transition in human coronary ECs

Because unbiased clustering produced three artery clusters in human but only two in mouse, we next investigated whether human hearts contained an artery cell state not present in mouse. This could occur because mouse and human arteries have some structural differences, for example, large conducting arteries in humans are on the surface rather than within the myocardium as in mouse (*Wessels and Sedmera, 2003*; *Kumar et al., 2005*; *Fernández et al., 2008*; *Sorop et al., 2020*). If these differences translated into an artery transcriptional state unique to human, we would expect one of the human artery clusters to not be represented in the mouse mapping. Instead, mouse cells mapped to all three human artery clusters (*Figure 6a*). There was also evidence indicating that human

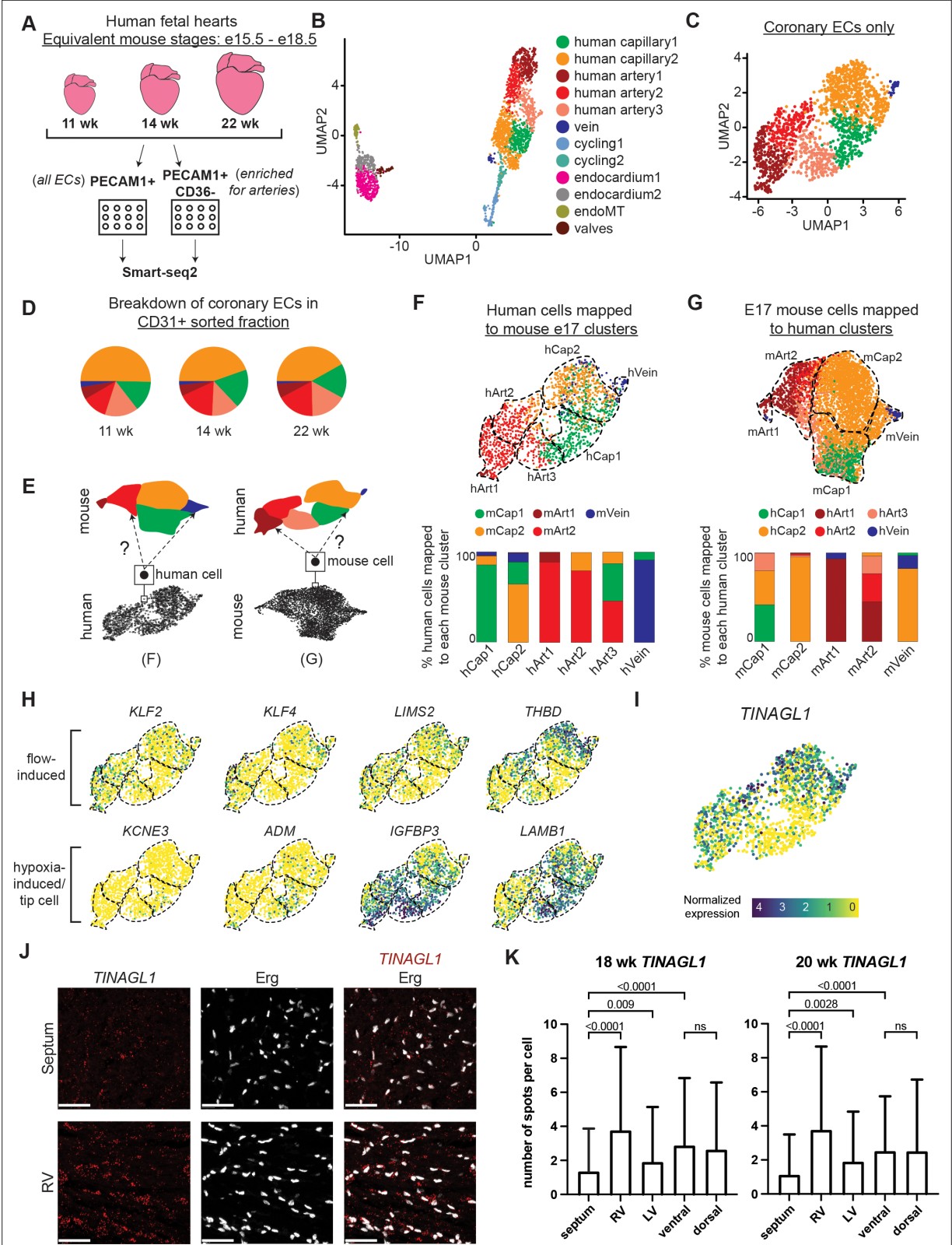

**Figure 5.** Single-cell RNA sequencing (ScRNAseq) of coronary endothelial cells (ECs) from human fetal hearts. (**A**) Overview of scRNAseq approach for three human fetal hearts. (**B and C**) Uniform Manifold Approximation and Projections (UMAPs) of all major *PECAM1+* EC subtypes collected (**B**) and the non-cycling coronary EC subset (**C**). (**D**) Pie charts showing the breakdown by cluster of human coronary ECs that were sorted as *PECAM1+* without additional enrichment. (**E**) Schematic of inter-species reference mapping. Individual cells from the human or mouse e17.5 datasets were assigned to the

*Figure 5 continued*

most similar mouse or human cluster, respectively. (**F and G**) Results from inter-species reference mapping based on shared gene expression, showing the mouse cluster that each human EC mapped to and the percentage breakdown of the mapping from each human cluster (**F**) and the converse comparison (**G**). Dashed lines show the borders of the previously defined human and mouse e17.5 coronary clusters. (**H**) UMAPs showing expression of selected flow-induced, hypoxia-induced and tip-cell genes in human coronary ECs. (**I**) UMAP showing expression of *TINAGL1* in human coronary ECs. Scale bar from (**I**) also applies to (**H**). (**J**) Section from 18-week human fetal heart showing in situ hybridization for *TINAGL1* with immunofluorescence for Erg. Scale bar = 50 µm. (**K**) Bar plot showing the mean number of *TINAGL1* RNA spots per cell detected in different regions of 18- and 20-week human fetal hearts. Error bars represent standard deviation.

The online version of this article includes the following figure supplement(s) for figure 5:

**Figure supplement 1.** Additional analysis of developing human coronary endothelial cells (ECs).

**Figure supplement 2.** Analysis of developing human coronary endothelial cells (ECs) separated by stage.

Art3 cells were in a less mature arterial state compared to human Art1 and Art2. This is because: (1) Art3 was unique among the human artery clusters in that a high proportion of Art3 cells matched mouse capillaries (*Figure 6A*). (2) Trajectory analysis with RNA velocity (*La Manno et al., 2018*), Slingshot (*Street et al., 2018*), and partition-based graph abstraction (PAGA) (*Wolf et al., 2019*) indicated a transition from human Art3 -> Art2 -> Art1 (*Figure 6B* and *Figure 5—figure supplement 2B*). (3) Comparing the fetal dataset with a publicly available adult dataset (*Litviňuková et al., 2020*) showed that almost no adult human artery ECs mapped to Art3, which would be predicted if Art3 were an immature developmental state (*Figure 6C* and *Figure 5—figure supplement 1F*). The direction of the arterial trajectory was determined by the arrows from the RNA velocity analysis, which provides this directional information as a consequence of comparing spliced (mature) to unspliced (immature) transcripts (*La Manno et al., 2018*). Additionally, this trajectory is supported by previous lineage analyses in mouse of a trajectory from capillaries to *Gja5-* arteries to *Gja5+* arteries (*Su et al., 2018*). Together, these data suggest that human hearts do not contain a dramatically unique artery transcriptional state when compared to mouse, but they do have an immature state (Art3) that matures into Art1 and Art2 in adults.

Reference mapping from the fetal to adult human datasets also revealed a notable reduction in Cap1 cells (28% of fetal capillary cells are Cap1 versus 6% of adult capillary cells) (*Figure 6C*). In mice, e17.5 Cap1 and Cap2 converged into a relatively homogenous capillary population in adult (*Figure 1I*). The small percentage of adult human coronary ECs mapping to the fetal Cap1 cluster**,** as well as the spatial overlap between cells mapping to Cap1 and Cap2 in the adult UMAP reduction, indicates that these developmental states related to flow and oxygenation also converge in humans.

We next performed trajectory analysis to investigate whether arteries develop through the differentiation of capillary ECs, which is the developmental pathway in mice (*Red-Horse et al., 2010*; *Su et al., 2018*). Two common methods for estimating trajectories, PAGA and Slingshot, identified connections between human Cap1 and Art3 and Cap2 and Art2 (*Figure 6B* and *Figure 5—figure supplement 1B*). RNA velocity suggested directionality going from the capillaries into arteries (*Figure 6B*). These two predicted capillary to artery transitions suggested that Cap1/Art3 and Cap2/Art2 may be differentiation trajectories occurring in two different locations in the heart, that is, septum versus ventricle walls. This is supported by the observation that several genes are specifically co-expressed in Cap1 and Art3 (including *CXCR7*, *MCAM*, and *TNFAIP8L1*, *PGF*) or in Cap2 and Art2 (including *TINAGL1*, *SLC9A3R2*, *SGK1*, *THBD*, *LIMS2*, *CALCRL*), some of which were shared with mouse (*Figures 5H and 6D*, and *Figure 5—figure supplement 1G*). From these data, we propose a model where, in both mouse and human, two distinct subtypes of capillary cells at different locations in the developing heart initially produce two subtypes of artery cells, one of which eventually matures into the other (*Figure 6E*).

## Characterization of human artery EC subpopulations

Since coronary artery disease is a leading cause of death and developmental information could suggest regenerative pathways, we next focused on the gene pathways present in developing human coronary arteries. As described above, unbiased clustering resulted in three artery states, each expressing known artery markers such as *GJA4* and *HEY1*, but also containing unique genes (*Figure 7A*). The *SCENIC* package (*Aibar et al., 2017*), which uses gene expression information to identify transcription factor 'regulons' present in cells, implicated *SOX17* as being strongly enriched in developing artery ECs (*Figure 7B*), which is consistent with previous reports on artery development (*Corada*

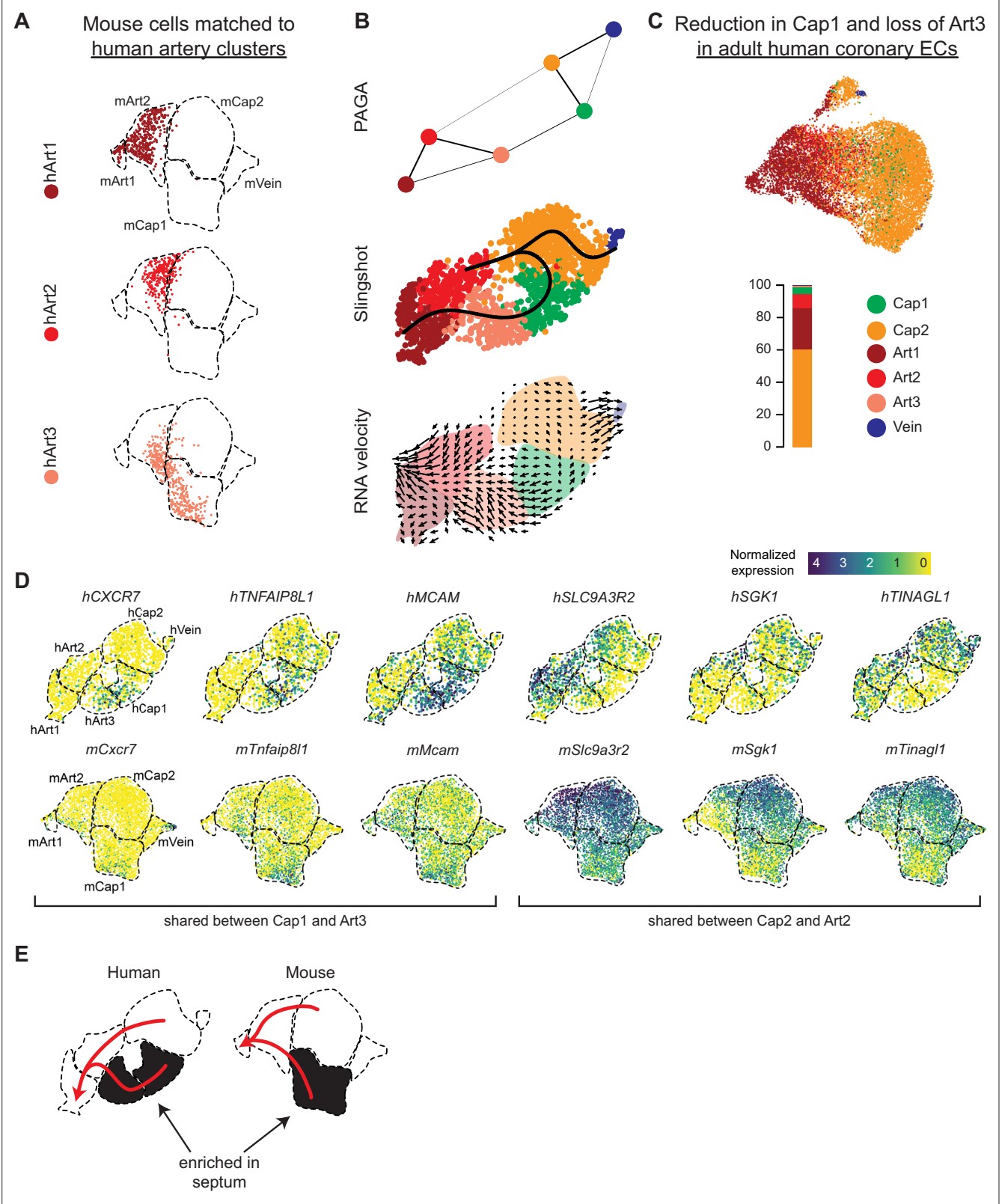

**Figure 6.** Trajectory analysis of developing human coronary arteries. (**A**) Reference mapping showed that e17.5 mouse endothelial cells (ECs) from *Figure 5g* were assigned to all three human artery subsets. (**B**) Trajectory analysis of human coronary ECs using partition-based graph abstraction (PAGA), Slingshot, and RNA velocity suggested that artery ECs are formed by capillary EC differentiation, as in mice. (**C**) Reference mapping adult human coronary ECs from a publicly available dataset to human fetal ECs showed that most mature cells match to Art1, Art2, or Cap2. (**D**) Uniform

*Figure 6 continued on next page*

*Figure 6 continued*

Manifold Approximation and Projections (UMAPs) showing expression of selected genes shared between hCap1 and hArt3 and hCap2 and hArt2, in both human and mouse. Previously defined clusters are outlined. (**E**) Schematic illustrating enrichment of septum ECs in mouse Cap1 and human Cap1 and Art3, as well as trajectories from capillary to artery in both human and mouse.

*et al., 2013*; *González-Hernández et al., 2020*). Transcription factors of potential importance that have not been previously implicated in artery development were *PRDM16* and *GATA2* (*Figure 7B*). Interestingly, the *IRF6* regulon was specific to the most mature population suggesting a potential role in artery maturation (*Figure 7C*). All of these regulons were similarly enriched in mouse artery cells (*Figure 7B and C*). We also identified several genes with strong expression patterns in human artery ECs that were not found in mouse (*Figure 7D* and *Table 4*). Interestingly, these included a GABA receptor, *GABBR2*, which was enriched in Art2, and a Glutamate receptor, *GRIA2*, which was is enriched in Art1. The human cells expressing *GABBR2* also co-expressed *SLC6A6*, a transporter that imports the GABBR2 ligand into cells (*Tomi et al., 2008*; *Figure 7D*). Finally, we localized different types of arteries in sections of human fetal hearts. In order to identify the artery subtypes, we used in situ hybridization for *GJA4* and *GJA5*. We found that *GJA5*-positive ECs, marking Art1, are in a small number of large arteries always covered with smooth muscle, while *GJA5*-negative/*GJA4*-positive ECs, marking Art2 and Art3, are numerous and in some cases not covered with smooth muscle (*Figure 7E and F*). This observation is consistent with the trajectory showed in *Figure 6B*, with the interpretation that the more mature *GJA5+* arteries in human are larger and more proximal than *GJA5-* arteries, as they are in mice. This supports the conclusion that Art1 represents artery ECs in larger, more proximal branches, and Art2 and Art3 are smaller arterioles (*Figure 7G*).

Cluster hArt1 (*GJA5+ GJA4+*) localizes to the largest arteries that are covered by mature SMMHC-positive smooth muscle (white arrows). Clusters hArt2 and hArt3 (*GJA4- GJA4+*) are smaller and can be either covered (yellow arrowheads) or not (red arrowheads) by smooth muscle. Scale bar = 100 μm. Scale bar from (E) also applies to (D). (G) Schematic of human coronary artery hierarchy.

## Discussion

Coronary ECs differentiate predominantly from the Endo and the SV. Although the spatial arrangement of these two lineages in the developing and adult mouse heart has been well characterized, it was previously unknown if either the distinct origin or localization of Endo- and SV-derived cells results in transcriptional or functional differences. Here, we used scRNAseq of lineage-traced ECs to address this question. At e12, when coronary vessels are just beginning to form, ECs transcriptionally segregated into two groups which are correlated with either Endo- or SV-specific expression patterns, and one of which is composed of only SV-lineage cells. However, at e17.5, during a phase of rapid growth and vascular remodeling, ECs segregated primarily based on being localized to either the septum or ventricular walls, the former of which expressed a genetic signature of experiencing low oxygen and blood flow. There was also differential gene expression between the dorsal and ventral walls of the developing heart. This could be due to differences in the timing and degree of their vascularization. We previously showed that coronary vessels are more numerous and provide more coverage on the dorsal side of the heart compared to the ventral side, at least until e15.5 (*Red-Horse et al., 2010*; *Chen et al., 2014*). Therefore, whatever environmental variables related to blood supply (including flow and hypoxia) distinguish the septum and the dorsal wall at e17.5 likely also cause the differences between the ventral and dorsal walls show in *Figure 3D–F*. In adult hearts, Endo- and SV-derived ECs cannot be distinguished either by gene expression or by their level of proliferation in response to IR injury. Altogether, these findings demonstrate that over the course of embryonic and post-natal development, coronary ECs from separate lineages converge both transcriptionally and functionally.

This result is relevant to future studies aiming to use developmental pathways to enhance regeneration in adult hearts. For example, it implies that approaches to stimulate regrowth of the vasculature after myocardial injury will affect all cells equally with regard to lineage, and that achieving vascular remodeling in adults may require replicating specific environmental cues and signals (especially those related to hypoxia and flow). The transcriptional similarity of adult Endo- and SV-derived cells helps explain the prior observation that mutant embryos whose coronary vasculature was derived primarily from the Endo due to compensation for loss of SV sprouting grew into phenotypically normal adults

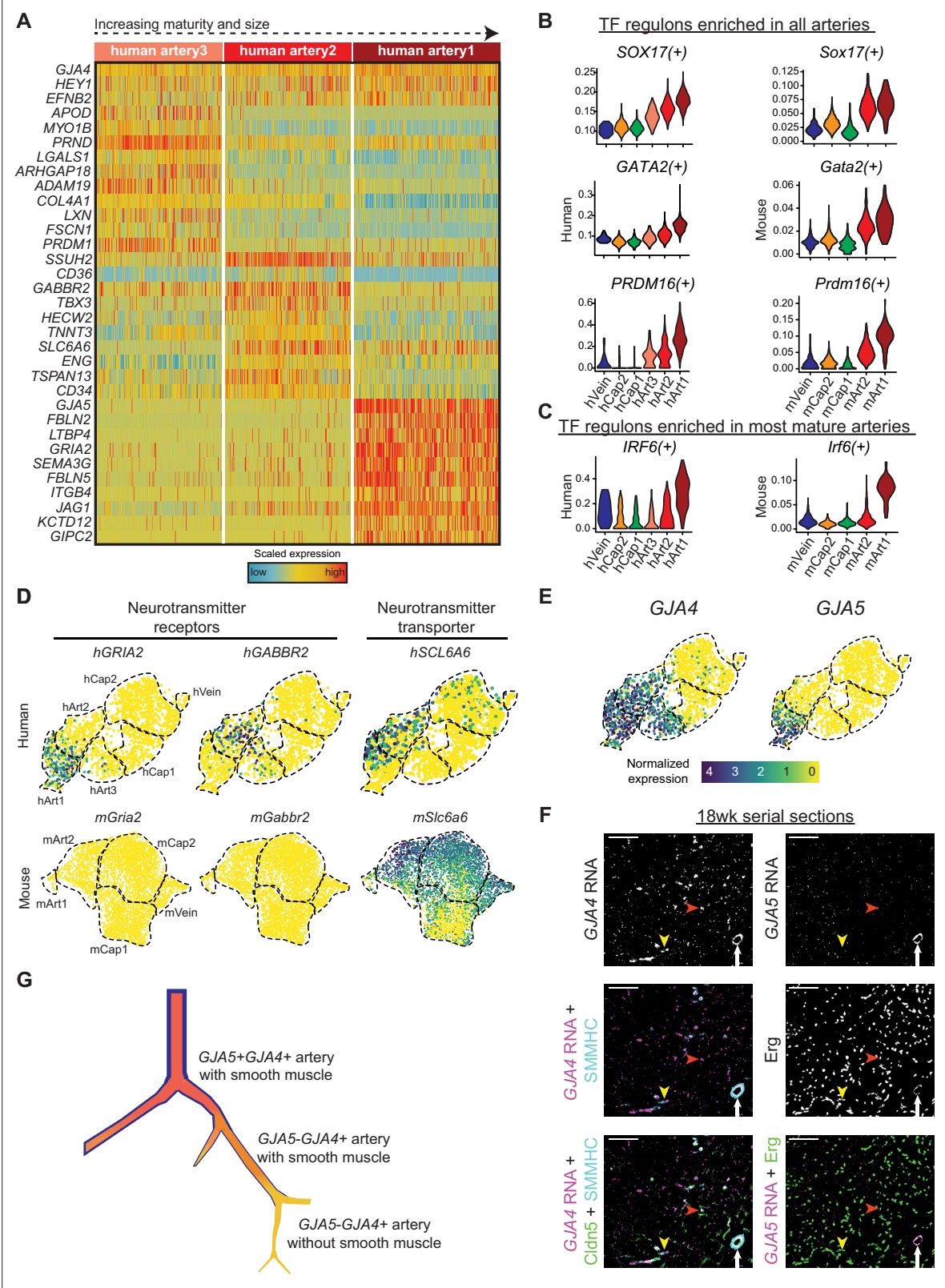

**Figure 7.** Gene expression in developing human coronary arteries. (**A**) Heatmap showing expression of selected genes enriched in human artery clusters. (**B and C**) Regulon scores from *SCENIC* analysis for TFs enriched in all human and mouse artery clusters (**B**) and for TFs enriched in human and mouse Art1 (**C**). (**D**) Human, but not mouse, developing coronary arteries expressed neurotransmitter receptors and their transporter. (**E**) *GJA4* and *GJA5* expression in human coronary endothelial cells (ECs). (**F**) Serial sections from 18-week human fetal heart showing in situ hybridization for the indicated mRNAs with immunofluorescence for the indicated proteins.

**Table 4.** Genes unique to human coronary endothelial cells (ECs).

| |
|---|
| SLC14A1 |
| GABBR2 |
| NRN1 |
| EPHA3 |
| ADMTSL1 |
| A2M |
| PRND |
| GRIA2 |
| KCNN3 |
| SERPINE2 |
| LPCAT2 |
| LGALS3 |
| APOA1 |
| SYNJ2 |
| OCIAD2 |
| PRICKLE2 |
| CTNND1 |
| IFITM2 |

despite developmental defects (*Sharma et al., 2017*). It also raises additional questions for investigation, namely, how is it that coronary ECs can over time lose the signatures both of their progenitors (between e12 and e17.5) and of their developmental 'home' (between e17.5 and adult)? Additional approaches, including ATAC-seq, could aid in addressing this question. The unexpected plasticity of these embryonic cells in their ability to become completely transcriptionally identical represents a potential model for efforts to stimulate faithful differentiation of very specific cell types from induced pluripotent stem cells or non-canonical progenitors (*Yamanaka, 2020*).

Another outstanding question addressed by this study is the degree of transcriptional similarity between developing mouse and human coronary ECs. Several groups have recently used scRNAseq to profile cell types in the fetal human heart, and identified individual genes that are enriched in either mouse or human (*Asp et al., 2019*; *Cui et al., 2019*; *Miao et al., 2020*; *Suryawanshi et al., 2020*). Here, our experiments enriched for ECs to enable a high-resolution comparison of this cellular compartment. The data showed that mouse transcriptional clusters specifically enriched in either septal or free wall cells are reproduced in our 11-, 14-, and 22-week human scRNAseq datasets, and many of the defining genes are conserved. Unlike in mouse, there is no apparent difference between ventral and dorsal gene expression patterns, indicating that the timing of vascularization might be more consistent throughout the developing human heart. Furthermore, the observation that adult human capillary ECs almost completely map to developmental Cap2 (*Figure 6C*) indicates that the hypoxic and low flow Cap1 is also resolved in adult human hearts as shown in mice. Although it is not possible to definitively identify the origins of human heart ECs using scRNAseq alone, the presence of similar cell states between mouse and human at these time points, as well as the convergence of adult capillary cells in both, lends confidence to the notion that coronary development generally follows the same progression in these two species.

Despite overall similarities in cell types, the scRNAseq analysis did reveal some interesting differences in gene expression between mouse and human, some of which may explain the anatomical differences in their vasculature. For instance, human Art1 and Art2 specifically expressed the glutamate receptor *GRIA2* and the GABA receptor *GABBR2*, respectively. It was previously demonstrated that exposure of ECs to GABA in vitro led to a reduction in response to inflammatory stimulus (*Sen et al., 2016*), and that mutations in brain ECs of a different GABA receptor, *Gabrb3*, resulted in defects in neuronal development in vivo (*Li et al., 2018*). These genes were not present in human adult coronary artery ECs. Further investigation may reveal an important role for GABA and glutamate signaling in human coronary development.

In summary, we have shown that in both mouse and human, phenotypically distinct lineage- and location-based cell states of coronary ECs converge in adults, and that embryonic lineage does not influence injury responses. This is a demonstration of the significant plasticity of the vasculature and

the influence environmental factors have in shaping heterogeneity, as well as the strength of mice as a model organism for human heart development.

# Materials and methods

**Key resources table**

| Reagent type (species) or resource | Designation | Source or reference | Identifiers | Additional information |
|---|---|---|---|---|
| Strain, strain background (*Mus musculus*) | Bmx$^{CreER}$ | Mouse Genome Informatics | MGI: 5513853; RRID: IMSR_TAC:14081 | |
| Strain, strain background (*Mus musculus*) | Rosa$^{tdTomato}$ | The Jackson Laboratory | Stock No: 007909; RRID: IMSR_JAX:007909 | |
| Strain, strain background (*Mus musculus*) | CD1 | Charles River Laboratories | Strain code: 022; RRID: IMSR_CRL:022 | |
| Biological sample (*Homo sapiens*) | Primary fetal heart tissue | Collected at Stanford from elective terminations | | |
| Antibody | Rat anti- APC/Cy7 Cd45 (rat monoclonal) | Biolegend | Cat #: 103116; RRID: AB_312981 | (1:50) |
| Antibody | Rat anti- APC Pecam1 (rat monoclonal) | Biolegend | Cat #: 102410; RRID: AB_312905 | (1:50) |
| Antibody | Rat anti- APC-Cy7 Ter119 (rat monoclonal) | Biolegend | Cat #: 116223; RRID: AB_2137788 | (1:50) |
| Antibody | Mouse anti- Pacific Blue CD235a (mouse monoclonal) | Biolegend | Cat #: 349107; RRID: AB_11219199 | (1:50) |
| Antibody | Mouse anti- FITC CD36 (mouse monoclonal) | Biolegend | Cat #: 336204; RRID: AB_1575025 | (1:50) |
| Antibody | Mouse anti- APC-Cy7 PECAM1 (mouse monoclonal) | Biolegend | Cat #: 303119; RRID: AB_10643590 | (1:50) |
| Antibody | Mouse anti- Pacific Blue CD45 (mouse monoclonal) | Biolegend | Cat #: 304021; RRID: AB_493654 | (1:50) |
| Antibody | Mouse anti- PerCP-Cy5.5 CD235a (mouse monoclonal) | Biolegend | Cat #: 349110; RRID: AB_2562706 | (1:50) |
| Antibody | Mouse anti-PerCP-Cy5.5 CD45 (mouse monoclonal) | Biolegend | Cat #: 304028; RRID: AB_893338 | (1:50) |
| Antibody | Rabbit anti-Erg (rabbit monoclonal) | Abcam | Cat #: ab92513; RRID: AB_2630401 | (1:200) |
| Antibody | Goat anti-Car4 (goat polyclonal) | R&D | Cat #: AF2414; RRID: AB_2070332 | (1:200) |
| Antibody | Rabbit anti-Smmhc (rabbit polyclonal) | Proteintech | Cat #: 21404–1-AP; RRID: AB_10732819 | (1:100) |
| Antibody | Mouse anti-Cldn5 (mouse monoclonal) | Invitrogen | Cat #: 35–2500; RRID: AB_2533200 | (1:200) |
| Peptide, recombinant protein | Collagenase IV | Worthington | Cat #: LS004186 | |
| Peptide, recombinant protein | Dispase | Worthington | Cat #: LS02100 | |
| Peptide, recombinant protein | DNase I | Worthington | Cat #: LS002007 | |
| Commercial assay or kit | RNEasy Mini Kit | Qiagen | Cat #: 74,104 | |
| Commercial assay or kit | iScript Reverse Transcription Supermix | Bio-Rad | Cat #: 1708840 | |
| Commercial assay or kit | Roche DIG RNA Labeling kit | Millipore Sigma | Cat #: 11175025910 | |

*Continued on next page*

*Continued*

| Reagent type (species) or resource | Designation | Source or reference | Identifiers | Additional information |
|---|---|---|---|---|
| Commercial assay or kit | RNAscope Multiplex Fluorescent V2 assay kit | Advanced Cell Diagnostics | Cat #: 323,100 | |
| Commercial assay or kit | Click-iT EdU Imaging kit | Thermo Fisher Scientific | Cat #: C10086 | |
| Chemical compound, drug | Tamoxifen | Sigma-Aldrich | Cat #: T5648 | |
| Chemical compound, drug | 4-OH Tamoxifen | Sigma-Aldrich | Cat #: H6278 | |
| Software, algorithm | Seurat v3 | https://doi.org/10.1016/j.cell.2019.05.031 | RRID: SCR_007322 | |
| Software, algorithm | bcl2fastq | illumina | RRID: SCR_015058 | |
| Software, algorithm | cutadapt 2.7 | https://doi.org/10.14806/ej.17.1.200 | RRID: SCR_011841 | |
| Software, algorithm | Cell Ranger v3.1.0 | 10× Genomics | RRID: SCR_017344 | |
| Software, algorithm | STAR v2.7.1a | https://doi.org/10.1093/bioinformatics/bts635 | RRID: SCR_004463 | |
| Software, algorithm | Subread v1.6.0 | https://doi.org/10.1093/nar/gkt214 | RRID: SCR_009803 | |
| Software, algorithm | heatmaply | http://dx.doi.org/10.1093/bioinformatics/btx657 | | https://github.com/talgalili/heatmaply (*Galili, 2021*) |
| Software, algorithm | biomaRt | 10.18129/B9.bioc.biomaRt | RRID: SCR_019214 | |
| Software, algorithm | PAGA | https://doi.org/10.1186/s13059-019-1663-x | | https://github.com/theislab/paga (*Thesis Lab, 2019*) |
| Software, algorithm | RNA velocity | 10.1038/s41586-018-0414-6 | RRID: SCR_018168; RRID: SCR_018167 | |
| Software, algorithm | Slingshot | https://doi.org/10.1186/s12864-018-4772-0 | RRID: SCR_017012 | |
| Software, algorithm | SCENIC | 10.1038/nmeth.4463 | RRID: SCR_017247 | |
| Software, algorithm | FIJI | doi:10.1038/nmeth.2019 | RRID: SCR_002285 | |
| Software, algorithm | QuPath | https://doi.org/10.1038/s41598-017-17204-5 | RRID: SCR_018257 | |
| Software, algorithm | Prism 8 | GraphPad Software | RRID: SCR_002798 | |
| Other | *TINAGL1* RNA probe | Advanced Cell Diagnostics | Cat #: 857221-C2 | |

## Mice

### Mouse strains

All mouse husbandry and experiments were conducted in compliance with Stanford University Institution Animal Care and Use Committee guidelines. Mouse lines used in this study are: *Bmx^CreER^* (*Ehling et al., 2013*), *tdTomato* (The Jackson Laboratory, B6.Cg-Gt(ROSA)26Sortm9(CAG-*tdTomato*)Hze/J, Stock #007909), and CD1 (Charles River Laboratories, strain code: 022).

### Breeding and tamoxifen administration

Timed pregnancies were determined by defining the day on which a plug was found as e0.5. For Cre inductions, tamoxifen (Sigma-Aldrich, T5648) was dissolved in corn oil at a concentration of 20 mg/ml and 4 mg was administered to pregnant dams using the oral gavage method on days e8.5 and e9.5 (*Figures 1A and 3D–E*, *Figure 1—figure supplement 1A*) or day e11.5 (*Figure 1b*). Combined injections of tamoxifen at e8.5 and e9.5 led to labeling of 94.44% of Endo cells and 3.61% of SV cells at e12.5. For the adult injury experiments, either 4 mg of tamoxifen or 1 mg of 4-OH tamoxifen (Sigma-Aldrich, H6278) was delivered on day e9.5 or e10.5 (*Figure 4A*, *Figure 1—figure supplement*

*1B*), respectively. The five *Bmx^CreER^-Rosa^tdTomato^* mice used for the adult 10× experiment were 13 weeks of age and all male. The 11 *Bmx^CreER^-Rosa^tdTomato^* adult mice used for the injury experiments were 12 weeks of age and all male (*Figure 4A*). The three CD1 adult mice used for quantification of EC proliferation in uninjured hearts were 10 weeks of age and all female (*Figure 4E, F and H*). The three CD1 adult mice used for Car4 staining were 6 weeks of age and all female (*Figure 3h*). Adult mice for the scRNAseq and injury experiments were obtained by harvesting litters at e18.5 and fostering pups with a different female who had given birth 0–4 days earlier.

## Human hearts

Under IRB approved protocols, human fetal hearts were collected for developmental analysis from elective terminations. Gestational age was determined by standard dating criteria by last menstrual period and ultrasound (*ACOG, 2009*). Tissue was processed within 1 hr following procedure. Tissue was extensively rinsed with cold, sterile PBS, and placed on ice in cold, sterile PBS before further processing as described below. Pregnancies complicated by multiple gestations and known fetal or chromosomal anomalies were excluded.

## scRNAseq protocol
### e12, e17.5, and adult mouse scRNAseq

*Bmx^CreER^-Rosa^tdTomato/tdTomato^* males were crossed to CD1 females, which were dosed with tamoxifen at e8.5 and e9.5 (e12 and e17.5) or with 4-OH tamoxifen at e10.5 (adult). Either early in the day on e12, or midday on e17.5, embryos were removed and placed in cold, sterile PBS. Forty-two Cre+ e12 embryos and 9 Cre+ e17.5 embryos were identified by their fluorescent signal and used for single-cell isolation. Five Cre+ adult males were identified by Cre amplification and used for single-cell isolation. Hearts were isolated and dissected to remove the atria and outflow tract, keeping the ventricles, SV, and valves (e12 and e17.5) or to remove the atria, outflow tract, and valves, keeping the ventricles (adult). Hearts were then dissociated in a 600 µl mix consisting of 500 U/ml collagenase IV (Worthington #LS004186), 1.2 U/ml dispase (Worthington #LS02100), 32 U/ml DNase I (Worthington #LS002007), and sterile DPBS with $Mg^{2+}$ and $Ca^{2+}$ at 37 degrees for 45 min and resuspended by pipetting every 5 min. Once digestion was complete, 5 ml of a cold 5% FBS in PBS mixture was added and the suspension was filtered through a 40 µm strainer. After further rinsing the strainer with 5 ml of 5% FBS/PBS, the cell suspension was centrifuged at 400 g at 4°C for 5 min. The cells were washed and resuspended once more in 1 ml 5% FBS/PBS. The following antibodies were added at the concentration of 1:50 and incubated on ice for 45 min: APC/Cy7 Cd45 (Biolegend #103116), APC Pecam1 (Biolegend #102410), APC/Cy7 Ter-119 (Biolegend #116223). DAPI (1.1 µM) was added to the cells immediately before FACS. Once stained, the cells were sorted on a Aria II SORP machine into 1.5 ml tubes. The gates were set up to sort cells with low DAPI, low Cd45 (hematopoietic cells), low Ter119 (erythroid cells), high Pecam1 (endothelial marker), and either high or low PE-Texas Red (*tdTomato* positive or negative). Compensation controls were set up for each single channel (PE-Texas Red, APC, APC/Cy7) before sorting the final cells. The samples were then submitted to the Stanford Genome Sequencing Service Center for 10× single-cell v3.1 3′ library preparation. For each stage, libraries from the *tdTomato*-positive and -negative samples were pooled and sequencing was performed on two lanes of a Illumina NovaSeq 6000 SP flow cell.

### Twenty-two week fetal human heart scRNAseq

The experiment was performed using the same procedure as the mouse samples unless noted here. The heart was kept in cold, sterile PBS. It was dissected to remove the atria, outflow tract, and valves, keeping only the ventricles. Dissociation was performed as described for the mouse samples except that multiple tubes of the 600 µl mix were used for each heart. The antibodies used for staining were: Pacific Blue CD235a (Biolegend #349107), FITC CD36 (Biolegend #336204), APC/Cy7 PECAM1 (Biolegend #303119), Pacific Blue CD45 (Biolegend #304021). The gates were set up to sort cells with low DAPI, low CD45 (hematopoietic cells), low CD235A (erythroid cells), high PECAM1 (endothelial marker), and low FITC. After staining each cell was sorted into a separate well of a 96-well plate containing 4 µl lysis buffer. Cells were spun down after sorting and stored at −80°C until cDNA synthesis. A total of 1920 PECAM1+ CD36 and PECAM1+ cells were sorted and processed for cDNA

synthesis. Cells were analyzed on the AATI 96-capillary fragment analyzer, and a total of 1382 cells that had sufficient cDNA concentration were barcoded and pooled for sequencing.

## Eleven- and Fourteen-week fetal heart scRNAseq

The experiment was performed using the same procedure as the 22-week heart unless noted here. The antibodies used for staining were PerCP/Cy5.5 CD235a (Biolegend #349110), PerCP/Cy5.5 CD45 (Biolegend #304028), FITC CD36 (Biolegend #336204), APC/Cy7 PECAM1 (Biolegend #303119). The gates were set up to sort cells with low DAPI, low CD45 (hematopoietic cells), low CD235A (erythroid cells), high PECAM1 (endothelial marker), and either low or high FITC. A total of 1824 PECAM1+ CD36-, PECAM1+ CD36+ and PECAM1+ cells from the 11-week heart and 1920 PECAM1+ CD36-, PECAM1+ CD36+ and PECAM1+ cells from the 14-week heart were sorted and processed for cDNA synthesis. A total of 1530 11-week and 1272 14-week cells that had sufficient cDNA concentration were barcoded and pooled for sequencing.

Synthesis of cDNA and library preparation for the fetal human heart cells was performed using the Smart-seq2 method as previously described (*Picelli et al., 2014*; *Su et al., 2018*). Libraries from the fetal human heart cells were part of a pool of samples that was sequenced on four lanes of a Illumina NovaSeq 6000 S4 flow cell.

## scRNAseq data analysis

### Processing of sequencing data

Raw Illumina reads for all datasets were demultiplexed and converted to FASTQ using *bcl2fastq* (Illumina). For human, sequencing adapter and PCR primer sequences were trimmed from reads using cutadapt 2.7 (*Martin, 2011*). For mouse, reads were aligned to GRCm38 Ensembl release 81 as well as *EGFP* and *tdTomato* sequences and a gene count matrix was obtained using Cell Ranger v3.1.0 (10× Genomics). For human, reads were aligned with STAR v2.7.1a (*Dobin et al., 2013*) to GRCh38 Ensembl release 98, and a gene count matrix was obtained using the *featureCounts* function of Subread v1.6.0 (*Liao et al., 2014*).

### Processing of count data

The majority of scRNAseq data analysis was performed using R and Seurat v3 (*Stuart et al., 2019*). Cells were deemed low-quality and excluded from downstream analysis if they expressed less than 1000 genes or if more than 10% of reads aligned to mitochondrial genes. A small number of cells were removed from the mouse e17.5 and adult *tdTomato*-negative samples which were expressing *tdTomato*. For all datasets, non-endothelial subtypes (e.g. blood and immune cells, cardiomyocytes, smooth muscle, fibroblasts) as well as a small number of lymphatic cells were removed. For the adult mouse, endocardial cells were removed, as well as a small cluster of cells enriched in dissociation-induced genes (e.g. *Hspa1a, Hspa1b, Socs3, Junb, Atf3*) (*van den Brink et al., 2017*). To obtain the subsets of vascular ECs shown in *Figure 1*, SV, valve, Endo, SV, and cycling cells were removed as shown in *Figure 1—figure supplement 2B*. Additionally, a cluster of cells with a lower gene count and higher mitochondrial percentage were removed from the e17.5 dataset. The cells used for the cell cycle analysis in *Figure 1—figure supplement 4* include all the cells used *Figure 1F* combined with the non-endocardial cycling cells shown in *Figure 1—figure supplement 2B*.

Count data from *Su et al., 2018*, and *Litviňuková et al., 2020*, were used to analyze gene expression in e12.5 mouse, and adult human hearts, respectively. From these datasets only vascular ECs were retained, excluding Endo, SV, valve endothelium, lymphatic endothelium, and non-EC types.

Normalization, variable feature selection, scaling, and dimensionality reduction using principal component analysis were performed using the standard Seurat v3 pipeline (*Stuart et al., 2019*). For the e17.5 and adult mouse datasets the technical variables genes per cell, reads per cell, and mitochondrial read percentage were regressed out in the *ScaleData* function. Following this, construction of a shared nearest neighbor graph, cluster identification with the Louvain algorithm (*Stuart et al., 2019*), and Uniform Manifold Approximation and Projection (UMAP) dimensionality reduction (*Becht et al., 2018*) were performed using the *FindNeighbors*, *FindClusters*, and *RunUMAP* functions in Seurat using the parameters listed below. Clustering resolution was determined individually for each dataset as the highest resolution at which every cluster expressed at least one unique marker gene:

*Figure 1—figure supplement 2A* (11,520 cells)—25 dimensions, Louvain resolution = 0.8
*Figure 1—figure supplement 2B* (12,205 cells)—25 dimensions, Louvain resolution = 0.8
*Figure 1* (436 cells)—25 dimensions, Louvain resolution = 0.6
*Figure 1* (4801 cells)—25 dimensions, Louvain resolution = 0.4
*Figure 1* (649 cells)—25 dimensions, Louvain resolution = 0.3
*Figure 2—figure supplement 2A* (356 cells)—20 dimensions, Louvain resolution = 1
*Figure 1—figure supplement 4A* (8495 cells)—25 dimensions, Louvain resolution = 0.4 (after regression), Louvain resolution = 0.6 (before regression)
*Figure 5* (2339 cells)—30 dimensions, Louvain resolution = 1.4
*Figure 5C* (1586 cells)—20 dimensions, Louvain resolution = 1
*Figure 6C* (13,47 cells)—50 dimensions, Louvain resolution = 0.8

For the human dataset, clustering with a resolution of 1 resulted in seven clusters: Art 1–3, Veins, Cap1, and two additional clusters which were eventually merged into Cap2. The reason for this merging is that one of these clusters was composed only of cells from the 22-week sample, and one of the top DEGs between these two clusters was *XIST* (as the 22-week sample was the only one expressing *XIST*, we concluded it was the only male sample). In addition, most of the other genes distinguishing these two clusters were dissociation-induced genes as described by *van den Brink et al., 2017*. Thus, it was determined that the difference between these two clusters was not biologically meaningful (mainly dissociation effects which were different between samples), and they were treated as one.

## Differential expression testing

Differential gene expression testing was performed with the *FindMarkers* and *FindAllMarkers* functions in Seurat using the Wilcoxon rank sum test. All differential genes were defined using parameters logfc.threshold = 0.3, min.pct = 0.2 and filtered for p-value < 0.001.

## Cell cycle regression

Cell cycle regression for *Figure 1—figure supplement 4* was performed using top 100 gene markers for the cycling clusters by p-value based on Wilcoxon rank sum test and the vars.to.regress parameter in the Seurat *ScaleData* function.

## Pearson correlation

Pearson correlation heatmaps in *Figure 2e–f* were created with the *heatmaply_cor* function from heatmaply (*Galili et al., 2018*).

## Dataset reference mapping

For cross-dataset mapping in *Figure 5*, the *getLDS* function in biomaRt (*Durinck et al., 2005*; *Durinck et al., 2009*) was used to identify every human gene that has a corresponding mouse gene and vice versa. Genes were only retained if they had a 1:1 mapping between human and mouse. The raw counts matrix for the human fetal data (all three sages pooled) was then filtered to include only these genes, and only the cells used in *Figure 5C*, and a new Seurat object was created from this count matrix. Similarly the raw count matrix for the e17.5 mouse data was filtered to include only these genes, and only the cells used in *Figure 1F and A*, and a new Seurat object was created from this count matrix. Standard normalization and scaling was performed in Seurat. To perform the mappings between datasets in *Figures 5E–G and 6C*, and *Figure 5—figure supplement 1F*, the Seurat functions *FindTransferAnchors* and *TransferData* were performed using 30 dimensions and canonical correlation analysis dimensionality reduction. The label transfer method employed is described in detail by *Stuart et al., 2019*. Briefly, diagonalized canonical correlation analysis is used for dimensionality reduction of both datasets, followed by L2 normalization. Then, a mutual nearest neighbors approach is used to identify pairs of cells ('anchors') between the two datasets that represent a similar biological state. Every cell in the query dataset is assigned an anchor in the reference dataset (with an associated anchor score), and the cluster label of the reference cell is assigned to the query cell.

The major limitation of this method as we applied it is that it is being used to compare datasets from different species that were generated using different methods (Smart-seq2 and 10×) and different sequencing depths. These differences as well as limitations associated with these methods impact the accuracy of the label transfer. For example, most of the mouse vein cells match to human

Cap2 rather than human veins (*Figure 5g*), likely because there are so few human vein cells while cells in the larger Cap2 population have a larger local neighborhood to strengthen their anchor score. In addition, the matching is limited to the clusters that are pre-annotated in each dataset (i.e., each human cell will match to its closest mouse cluster, even if there is not a true biological correlate). Finally, genes used in the mapping were limited to those that had a 1:1 homology mapping between mouse and human, meaning that some information was eliminated before label transfer.

## Trajectory analysis

Trajectory analyses shown in *Figure 6b* and *Figure 5—figure supplement 2b* and were performed with PAGA (*Wolf et al., 2019*) (filtered for edge weight greater than or equal to 0.14), RNA velocity (*La Manno et al., 2018*) (using the python function *veloctyto run-smartseq2*, followed by the R package velocyto.R with the function *show.velocity.on.embedding.cor* with fit.quantile = 0.05, grid.n = 20, scale = 'sqrt', arrow.scale = 3, and n = 50–100), and Slingshot (*Street et al., 2018*) (using the *slingshot* function with Cap2 as the starting cluster and stretch = 1).

## Transcription factor enrichment

Transcription factor enrichment was performed with SCENIC (*Aibar et al., 2017*) using the *pyscenic* functions and the recommended pipeline (*Van de Sande et al., 2020*). The loom file output from SCENIC was then imported into Seurat, and the *FindAllMarkers* function with the Wilcoxon rank sum test was used to identify differential regulons between clusters.

## Immunofluorescence and imaging

### Tissue processing and antibody staining

E17.5 embryos were dissected in cold 1× PBS and fixed in 4 % PFA for 1 hr at 4°C, followed by three 15 min washes in PBS. Hearts were then dissected from the embryos. Adult mouse hearts were dissected and fixed in 4% PFA for 4–5 hr at 4°C, followed by three 15 min washes in PBS. Hearts were dehydrated in 30% sucrose overnight at 4°C, transferred to OCT for a 1 hr incubation period, and frozen at –80°C. For each heart, the whole ventricle was cut into 20-μm-thick sections which were captured on glass slides. Staining was performed by adding primary antibodies diluted in 0.5% PBT (0.5% Triton X-100 in PBS) with 0.5% donkey serum to the sections and incubating overnight at 4°C. The following day the slides were washed in PBS 3 times for 10 min followed by a 2 hr room temperature incubation with secondary antibodies, three more 10 min washes, and mounting with Fluoromount-G (SouthernBiotech #0100–01) and a coverslip fastened using nail polish. Human fetal hearts were fixed in 4% PFA for 24–48 hr at 4°C, followed by three 15 min washes in PBS. The hearts were sequentially dehydrated in 30%, 50%, 70%, 80%, 90%, and 100% ethanol, washed three times for 30 min in xylene, washed several times in paraffin, and finally embedded in paraffin which was allowed to harden into a block. For each heart, the whole ventricle was cut into 10-μm-thick sections which were captured on glass slides.

### In situ hybridization for GJA4 and GJA5

RNA was isolated from a 23-week human fetal heart using Trizol-based dissociation followed by the RNEasy Mini Kit (Qiagen #74104). cDNA was created from this RNA using the iScript Reverse Transcription Supermix (Bio-Rad #1708840). Primers used to amplify *GJA4* are 5'-AAACTCGAGAAGATCT CGGTGGCAGAAGA-3' and 5'-AAATCTAGACTGGAGAGGAAGCCGTAGTG-3'. Primers used to amplify *GJA5* are 5'-AAACTCGAGAATCAGTGCCTGGAGAATGG-3' and 5'-AAATCTAGATGGTCCA TGGAGACAACAGA-3'. Digoxin-linked probes were transcribed using the Roche DIG RNA Labeling Kit (Millipore Sigma #11175025910). In situ hybridization was performed as previously described (*Koop et al., 1996*) with a modification to develop the fluorescent signal. Briefly, after hybridization, sections were incubated overnight at 4°C with anti-DIG POD (Millipore Sigma #11207733910). The next day, sections were washed 4 × 1 hr in 1× MABT. Finally, sections were washed for 3 × 10 min with 0.1 M borate buffer pH 8.5 and stained with bench-made tyramide (*Vize et al., 2009*).

## In situ hybridization for TINAGL1

In situ hybridization was performed using the RNAscope Multiplex Fluorescent V2 assay (Advanced Cell Diagnostics #323100), with probe for human *TINAGL1* (Advanced Cell Diagnostics #857221-C2) and OPAL 570 fluorophore (Akoya #FP1488001KT), following the manufacturer's protocol.

## Microscopy and image processing

Images were captured on a Zeiss LSM-700 confocal microscope. For each experiment, littermate embryos were stained together and all samples were imaged using the same laser settings. For each experiment, laser intensity was set to capture the dynamic range of the signal. Images were captured using Zen (Carl Zeiss) and processed using FIJI (NIH) and Illustrator (Adobe). Any changes to brightness and contrast were applied equally across the entire image. All mouse imaging experiments were performed with at least three individual samples.

## Antibodies

The following primary antibodies were used anti-ERG (1:200; Abcam, ab92513), anti-Car4 (1:200; R&D, AF2414), anti-Smmhc (1:100; Proteintech, 21404–1-AP), anti-Cldn5 (1:100; Invitrogen, 35–2500). Secondary antibodies were Alexa Fluor-conjugated antibodies (488, 555, 633) from Life Technologies used at 1:250.

## Quantification

Quantification of Car4 (*Figure 3d–e*, *Table 3*) and EdU (*Figure 4e–h*) was performed using the Cell-Counter plugin in FIJI. For e17.5 embryos, Car4, Erg, and *tdTomato* were quantified in five ROIs in each of three sections from each of three hearts. The ROIs were 510 μm × 190 μm and were positioned to maximize coverage of the septum, left ventricular wall, right ventricular wall, dorsal wall, and ventral wall. For each heart, counts were combined across the three sections. For adult hearts, Car4 and Erg were quantified in two ROIs of 600 μm × 600 μm, one in the septum and one in the left ventricle.

For the lineage comparison in adult injured hearts (*Figure 4c–f*), EdU, Erg, and *tdTomato* were quantified in two ROIs for each of 11 hearts at the level of the stitch (quantification from two ROIs was averaged), and in one ROI for each of seven hearts at the apex. ROIs were 600 μm × 600 μm and were chosen to be in the region of the section with the greatest density of EdU staining, and whenever possible, to span portions of both the inner and outer myocardial wall. For uninjured controls for the lineage comparison, one ROI of 600 μm × 600 μm was chosen in both the middle of the myocardial wall of the left ventricle, and at the apex, from each of three uninjured 6-week female CD1 mouse hearts. *Bmx^CreER^-Rosa^tdTomato^* was observed to label ECs in large arteries in adult tissues even without tamoxifen administration (Red-Horse Lab, unpublished data). To compare proliferation in the inner and outer wall of the adult heart (*Figure 4f*), one ROI each of 600 μm × 600 μm were chosen in a portion of the injured area (areas with high density of EdU staining, as indicated in *Figure 4b*) overlapping with the inner or outer myocardial wall, in three injured hearts. In addition, one ROI each of 600 μm × 600 μm was chosen adjacent to the injured area, and overlapping with the inner or outer myocardial wall, in three injured hearts. Since the developmental origin of adult arteries cannot be determined using *Bmx^CreER^*, large arteries were excluded from all ROIs during quantification for the lineage comparison after injury. However, we were able to use tdTomato labeling to identify arteries in the injury area for the quantification of artery proliferation. For each of 10 hearts at the level of the stitch, and for each of four hearts at the apex, large arteries labeled with tdTomato were identified in the injury area, and the number of Erg+ and Erg+ Edu + cells were counted. These were compared to counts of Erg+ and EdU + cells in arteries identified in the left ventricles (three arteries each) and apex (two arteries each) of three uninjured 6-week female CD1 mouse hearts, with the arteries being identified by Smmhc staining.

For human *TINAGL1*, ROIs were captured at 40× magnification from one section each from an 18-week and a 20-week human fetal heart. For each section, four ROIs of 320 μm × 320 μm were chosen from the septum and left ventricle, and three ROIs of the same size were chosen from the right ventricle, ventral wall, and dorsal wall. Automatic detection of Erg+ nuclei and probe spots were

performed in QuPath using the default parameters with the following exceptions: cellExpansionMicrons = 0.1, subcellular detection threshold = 100. Measurements shown in *Figure 5k* represent the values of 'Num spots estimated'.

Graphs in *Figures 3–5* and *Figure 3—figure supplement 1* were made in Prism 8.

## Statistics

In *Figure 4e*, paired t-tests were used to compare Endo-derived and SV-derived EC proliferation, and Welch's t-test was used to compare Endo-derived or SV-derived EC proliferation with the uninjured control. Unpaired t-tests were used for *Figures 3e and 4f*. Unpaired Welch's t-tests were used for *Figure 4h*. For *Figure 5k*, comparisons were made using one-way ANOVA with Holm-Sidak's multiple comparisons test.

## IR injury experiment

### Surgery

$Bmx^{CreER}$-$Rosa^{tdTomato/tdTomato}$ males were crossed to CD1 females, which were dosed with tamoxifen e9.5 or with 4-OH tamoxifen at e10.5. In addition to being pharmacologically less toxic compared to Tam, 4-OHT is more potent at inducing Cre, given its stronger affinity for the ER domain in *CreER* strain (*Robertson et al., 1982*; *Katzenellenbogen et al., 1984*; *Cardoso et al., 2003*). By administering 4-OHT 1 day later than Tam, Cre was induced in all animals at approximately the same developmental time regardless of treatment. Pups were dissected from the pregnant dams at day e18.5 and fostered as described above. IR was performed in 12-week-old mice by the Stanford Murine Phenotyping Core (SMPC) that is directed by Dr Dan Bernstein. To summarize, mice were anesthetized using isoflurane and placed on a rodent ventilator to maintain respiration before opening the chest cavity. The LAD coronary artery was ligated with a 8.0 silk suture and resulting ischemia of the myocardium was verified by blanching the left ventricular wall. After 40 min, the suture was removed around the LAD, allowing for the reperfusion of downstream myocardium. To end the procedure, the chest was closed, and post-operative analgesia was administered to the mice.

### In vivo proliferation assay

To assess EC proliferation after IR, 5-ethynyl-2′-deoxyuridine (EdU) (Thermo Fisher Scientific, cat. #E10415) was diluted 2.5 mg/mL in sterile PBS and injected intraperitoneally on days 3 and 4 post-injury at a dose of 10 μl/ g body weight. Hearts were harvested from sacrificed animals on day 5 post-injury. Cryosectioning and antibody staining for Erg was performed as described above. To detect endothelial proliferation, the protocol for Click-iT EdU Imaging (Thermo Fisher Scientific, cat. #C10086) was carried out according to manufacturer's instructions.

## Acknowledgements

KR is supported by the NIH (R01-HL128503). RP is supported by an AHA graduate fellowship. Sequencing of the adult mouse and fetal human datasets was funded by the Chan Zuckerberg Biohub. We thank Ralf Adams for sharing the $Bmx^{CreER}$ mouse line. We thank the Stanford Family Planning Clinic and Purnima Iyer Narasimhan for assistance with tissue procurement. We thank Gavin Sherlock for access to equipment needed for single-cell library preparation. We thank all members of the Red-Horse lab for technical and intellectual support. We thank Rahul Sinha, Anshul Kundaje, and Laksshman Sundaram for discussion and advice about scRNAseq technique and analysis. We thank Biafra Ahanonu for discussion and advice about figure and manuscript preparation. We thank members of the Stanford Genome Sequencing Services Center which is supported by NIH Grant # 1S10OD020141-01. VDW is supported by the H&H Evergreen Fund.

## Additional information

### Funding

| Funder | Grant reference number | Author |
|---|---|---|
| National Heart, Lung, and Blood Institute | R01-HL128503 | Kristy Red-Horse |
| American Heart Association | Predoctoral fellowship | Ragini Phansalkar |
| Chan Zuckerberg Biohub | Funding for sequencing | Kristy Red-Horse |

The funders had no role in study design, data collection and interpretation, or the decision to submit the work for publication.

### Author contributions

Ragini Phansalkar, Conceptualization, Data curation, Formal analysis, Investigation, Methodology, Project administration, Validation, Visualization, Writing – original draft, Writing – review and editing; Josephine Krieger, Formal analysis, Investigation, Writing – original draft; Mingming Zhao, Investigation, Methodology; Sai Saroja Kolluru, Robert C Jones, Stephen R Quake, Resources; Irving Weissman, Daniel Bernstein, Resources, Supervision; Virginia D Winn, Data curation, Investigation, Project administration, Resources, Supervision; Gaetano D'Amato, Conceptualization, Data curation, Investigation, Methodology, Project administration, Supervision, Writing – original draft; Kristy Red-Horse, Conceptualization, Funding acquisition, Methodology, Project administration, Resources, Supervision, Writing – original draft, Writing – review and editing

### Author ORCIDs

Ragini Phansalkar (ID) http://orcid.org/0000-0003-3014-1915
Robert C Jones (ID) http://orcid.org/0000-0001-7235-9854
Virginia D Winn (ID) http://orcid.org/0000-0003-1136-2907
Gaetano D'Amato (ID) http://orcid.org/0000-0003-2640-5826
Kristy Red-Horse (ID) http://orcid.org/0000-0003-1541-601X

### Ethics

All animals were handled according to approved institutional animal care and use committee (IACUC) protocols (#26923 and #33123) of Stanford University. Animals were monitored regularly by the researchers as well as by veterinary services technicians and ACLAM board certified veterinarians. Euthanasia was performed in accordance with the Panel on Euthanasia of the American Veterinary Medical Association. During the ischemia-reperfusion injury, animals were anesthetized with isoflurane, and received buprenorphine after the procedure to alleviate discomfort.

### Decision letter and Author response

Decision letter https://doi.org/10.7554/eLife.70246.sa1
Author response https://doi.org/10.7554/eLife.70246.sa2

## Additional files

### Supplementary files

• Transparent reporting form

### Data availability

Gene count matrices from the mouse and human single-cell RNA sequencing and FASTQ reads from the mouse single-cell sequencing generated in this study have been deposited on GEO with accession numbers: GSE213274 (mouse only), GSE213275 (human only), GSE213276 (mouse and human). The FASTQ reads from the human single-cell RNA sequencing will be made available upon request.

The following datasets were generated:

| Author(s) | Year | Dataset title | Dataset URL | Database and Identifier |
|---|---|---|---|---|
| Phansalkar R, Krieger J, Zhao M, Kolluru SS, Jones RC, Quake SR, Weissman I, Bernstein D, Winn VD, D'Amato G, Red-Horse K | 2022 | Coronary blood vessels from distinct origins converge to equivalent states during mouse and human development (Mouse) | https://www.ncbi.nlm.nih.gov/geo/query/acc.cgi?acc=GSE213274 | NCBI Gene Expression Omnibus, GSE213274 |
| Phansalkar R, Krieger J, Zhao M, Kolluru SS, Jones RC, Quake SR, Weissman I, Bernstein D, Winn VD, D'Amato G, Red-Horse K | 2022 | Coronary blood vessels from distinct origins converge to equivalent states during mouse and human development (Human) | https://www.ncbi.nlm.nih.gov/geo/query/acc.cgi?acc=GSE213275 | NCBI Gene Expression Omnibus, GSE213275 |
| Phansalkar R, Krieger J, Zhao M, Kolluru SS, Jones RC, Quake SR, Weissman I, Bernstein D, Winn VD, D'Amato G, Red-Horse K | 2022 | Coronary blood vessels from distinct origins converge to equivalent states during mouse and human development | https://www.ncbi.nlm.nih.gov/geo/query/acc.cgi?acc=GSE213276 | NCBI Gene Expression Omnibus, GSE213276 |

The following previously published datasets were used:

| Author(s) | Year | Dataset title | Dataset URL | Database and Identifier |
|---|---|---|---|---|
| Litviukov M | 2020 | Cells of the Adult Human Heart | https://cellgeni.cog.sanger.ac.uk/heartcellatlas/data/hca_heart_vascular_raw.h5ad | European Nucleotide Archive, PRJEB39602 |
| Su T | 2018 | Single-cell analysis of early progenitor cells that build coronary arteries | https://github.com/gmstanle/coronary-progenitor-scRNAseq | Github, Github |

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
