## [Editor Report]

The authors largely improved the paper with additional analyses and extensive revisions. By employing scRNA-seq analyses, they elegantly have dissected the endothelial cell (EC) heterogeneity of cardiac blood vessels across development in mouse and human. They convincingly demonstrate that the EC heterogeneities of cardiac blood vessels are sequentially governed by the progenitor sources and environmental cues during initial and late development. Moreover, they show that these ECs become homogeneous in adult. They also claim that human fetal hearts take on generally a similar path for establishment of cardiac blood vessels. Overall, this study is novel and intriguing.

---

## [Decision Letter]

**Decision letter after peer review:**

Thank you for submitting your article "Coronary blood vessels from distinct origins converge to equivalent states during mouse and human development" for consideration by *eLife*. Your article has been reviewed by 3 peer reviewers, including Gou Young Koh as Reviewing Editor and Reviewer #1, and the evaluation has been overseen by Edward Morrisey as the Senior Editor. The following individual involved in review of your submission has agreed to reveal their identity: Victoria L Bautch (Reviewer #2).

In the light of the comments made by the reviewers, we decided to offer a "revision" prior to publishing your manuscript in *eLife*.

The comments of all three reviewers are in good agreement. While the reviewers found this study is novel and interesting, they raised concerns about the immature interpretations by under-clustering or superficial clustering in the substantial parts of sc-RNA data. The authors are required to carefully address the comments point-by-point in a data-driven manner or with further analyses or discussions. Specifically, the authors are encouraged to pay attention to the major comments 1-3 of reviewer 1 and comments 1-3 of reviewer 3. If necessary, please provide the reasons for not implementing the suggested changes.

I believe the authors could revise the manuscript successfully given their expertise, but please let us know if it will take more than 3 months.

*Reviewer #1 (Recommendations for the authors):*

1. The resolution of clustering of e17.5 dataset is ambiguous. It seems there are significant differences between septal and non-septal Cap1 ECs, but it is unclear why they were clustered into a single Cap1 cluster. Likewise, it seems there is a significant lineage-based distinction within Cap1 ECs in the heatmap (Figure 2a), all of which could be caused by under-clustering. Please provide an adequate rationale for the cluster identification on the e17.5 dataset.

2. The results shown in Figure S9 and Figure 3c-e and the interpretations in line 247-267 are insufficient to support the author's claim regarding the source of heterogeneity in e17.5 Caps. Authors need to provide more definitive evidence to exclude possible lineage-based contribution for heterogeneity in the e17.5 caps.

3. In Figure 1i-k, authors claim that Cap ECs of adult heart are relatively homogeneous. If so, how were they identified as distinct clusters? On the other hand, analysis of a previous study (Kalucka et al., Cell 2020) showed clear distinction between Cap1 and Cap2. Please describe the procedures for cluster identification in detail and explain the discrepancies between two datasets.

4. The authors are required to describe in detail about the label transfer method and its limitations to follow up. It was employed for the comparison of human and mouse datasets, but this method could raise a biased characterization of the cell types. Therefore, the authors are strongly recommended to confirm the EC heterogeneity in human fetal hearts by IHC or ISH.

5. The inference of developmental pathway using the trajectory analysis is not straightforward. In Figure 6b, the trajectory analysis predicts development from Art3 to Art2 and Art1, which does not match with their localizations as shown in Figure 7f-g. Authors need to describe how they made the direction of the flow in the trajectory analysis.

*Reviewer #2 (Recommendations for the authors):*

Overall, this is a very rigorous and complete study of the expression and functional potential of the two lineages that contribute to coronary EC. The range of the work is impressive, with scRNA seq data complemented by spatial localization and injury model, and by cross-referencing with new human scRNA seq datasets. Good use was made of publicly available datasets as well. The conclusion that coronary EC have transcriptional profiles that reflect location rather than original source is well-supported, as is the finding that they do not appear to have functional differences in response to injury. The data analysis of regulons is interesting; it would also be interesting to consider whether there might be an epigenetic memory of lineage source – perhaps this could be part of the discussion, now that the initial gene expression lineage differences are so well-documented in this study.

*Reviewer #3 (Recommendations for the authors):*

The authors claim is well supported by their data. scRNAseq and histological examination clearly support their idea "convergent differentiation" of coronary ECs, although the results shown in the present study are descriptive. There are several points that should be addressed before publication.

1. According to the previous reports by several groups including the authors' group, the SV-derived cells and the Endo-derived cells contribute to the vessels in the outer myocardial wall and inner myocardial wall. Now it is striking difference of the gene expression of ECs between ventral and dorsal wall of myocardium (Figure 3). Neither blood flow nor ischemia might not account for this regional difference at e17.5. If possible, the authors are encouraged to explain or discuss the difference of dorsal and ventral wall, although both are outer layer of myocardium.

2. To compare the gene expression of ECs between mouse and human, the authors used the Seurat label Transfer workflow to reference map each human cell to its closest mouse cluster (Figure 5e-g). Indeed, human Cap1 might be a cluster enriched of septal cells with less blood flow. Therefore, the authors claim that similar environment (ischemia and flow) between mouse and human contributes to clustering of coronary ECs.

Given the methods described for human heart sections (line 707-711), human tissues are available for examining expression of genes listed in Figure S10c or 10d as that of mice (Kcne3) was examined in Figure 3f. Histological examination of gene or protein expression using human hearts to investigate where genes liste in Figure S10 would support the authors' claim.

3. Related to 1 and 2, the distinct gene expression between inner and outer myocardial wall and between dorsal and ventral wall found in mouse hearts could be examined by immunohistochemical or in situ hybridization analyses using human embryonic hearts. These experiments strengthen the conclusion of comparable "convergent differentiation of coronary Cap ECs" between mouse and human.

---

## [Author Response]

Reviewer #1 (Recommendations for the authors):1. The resolution of clustering of e17.5 dataset is ambiguous. It seems there are significant differences between septal and non-septal Cap1 ECs, but it is unclear why they were clustered into a single Cap1 cluster.

We agree with the reviewers that it is best to have a standardized and clear rationale for choosing clustering resolution. We therefore reanalyzed all data choosing a standard criteria aimed at guarding against over-clustering into biologically less meaningful groups. Specifically, we increased the resolution until the additional clusters emerging no longer expressed unique genes, but instead separated out merely due to small differences in the levels of a few genes. For instance, when comparing expression patterns of top differentially expressed genes (DEGs) between e17.5 Cap1 and Cap2 derived from a resolution of 0.4 to the DEGs between additional clusters that emerged at higher resolution (0.5 and above), we observe a shift from distinctive expression differences to gradients of expression (Author response image a). Note that re-analyzing with this criteria did not change our original clustering. We clarified our use of this criterion for selecting resolution at page 10, lines 131-132 in the text and page 59, lines 686-688 in the *Methods*.

To address the specific comment on Cap1 not separating into two clusters at resolution 0.4 due to low resolution clustering, we observed clusters in 7 additional resolutions up to 1.1. In all cases, no clusters overlapping the gene expression patterns we proposed were septum vs. non-septum emerged (Author response image 1) . We do not know why the clustering algorithms do not pull out the septal cluster, but it could be because we used a combination of gene expression, protein staining, and lineage information, (Figure 3b-f, Figure 3—figure supplement 2), which the clustering algorithms do not take into account. This point has been added to the text at page 27, lines 263-265.

**Author response image 1. sa2fig1:** . (A) UMAP plots showing clustering of e17.5 dataset at multiple resolutions up to 0.6 as well as expression of the top 5 DEGs (by average log-fold change) between newly divided clusters. The clusters being compared for each resolution are indicated with a dashed line. (B) UMAP plots showing clustering of e17.5 dataset at multiple resolutions up to 1.1. The dashed line indicates the region of proposed septal cells based on lineage, protein staining and gene expression.

Likewise, it seems there is a significant lineage-based distinction within Cap1 ECs in the heatmap (Figure 2a), all of which could be caused by under-clustering. Please provide an adequate rationale for the cluster identification on the e17.5 dataset.

We agree that there does look like a lineage distinction on the heatmap in Cap1. However, examining these genes on the UMAP revealed that this was because Cap1 happened to contain the septum cells, which are mostly from the Endo (Chen et al., 2014) (Figure 3c), and the data in subsequent analyses (Figure 3—figure supplement 2) indicated that the septum imparts a location-specific effect on transcription. Thus, what looks like a lineage difference on the heatmap on further inspection was actually a difference of location because most septal cells are from the endocardium.

The key to concluding that the septal gene expression patterns were due to location rather than lineage was that the Endo cells outside the septum tended to downregulate these genes and the SV cells inside the septum upregulated them. To demonstrate this point, we revised Figure 3—figure supplement 2 to show gene expression plots of selected Endo-specific genes from Figure 2a split into Endo- and SV-derived samples. We quantified the percentage of cells expressing these genes in: 1. Endo-derived cells in the septum, 2. SV-derived cells in the septum, 3. Endo-derived cells outside the septum, and 4. SV-derived cells outside the septum (revised Figure 3—figure supplement 2b). If the expression of these genes was primarily influenced by lineage, we would expect to see them in high percentages of Endo-derived cells both inside and outside the septum. Conversely, if expression was primarily influenced by location, we would expect to see them in high percentages of both Endo- and SV-derived cells in the septum, but lower outside the septum. For example see .Author response image 2

The overriding patterns of the genes appearing enriched in Cap1 Endo-CVs in Figure 2a were: 1. None of the genes were expressed in high enough percentages to pass our significance threshold (between 0-35% of cells), and 2. There was a decreased expression in Endo- *and* SV-derived cells outside of the septum, supporting the location hypothesis (revised Figure 3—figure supplement 2b). Only two genes had a pattern reminiscent of a predicted lineage effect—*Gucy1b3* and *Hand2*— (revised Figure 3—figure supplement 2b) but they were expressed in such a low percentage of cells. We therefore concluded that our data supported a model where there was a heavy influence of location, but a potentially minor retention of two genes from the progenitor state in a small subset of cells. We noted this in the text at pages 27-29, lines 274-283.

2. The results shown in Figure S9 and Figure 3c-e and the interpretations in line 247-267 are insufficient to support the author's claim regarding the source of heterogeneity in e17.5 Caps. Authors need to provide more definitive evidence to exclude possible lineage-based contribution for heterogeneity in the e17.5 caps.

At the onset of our study, we predicted that if lineage information was retained in developing CVs, it would be a source of transcriptional heterogeneity and result in cell clustering that correlated with lineage or a retention of a significant number of lineage-specific genes. We observed this pattern at e12. However, our data led us to conclude that location, and not lineage, is the predominant source of heterogeneity in e17.5 capillaries because: (1) The differences between Cap1 and Cap2 are the major axis of heterogeneity based on unbiased clustering, and these two clusters contain many cells of both lineages, and (2) The Cap2 marker, *Car4*, localized Cap1 and Cap2 to distinct regions of the heart, but was expressed to a similar degree in Endo- and SV-derived cells at any particular location. We included a clearer, less complicated presentation of this second point in revised Figure 3e (the original panel from Figure 3e was moved to Figure 3—figure supplement 1b). Our understanding is that the reviewer suggests more data beyond these results, which we included and which are described below.

We performed additional analyses with the prediction that significant lineage-related heterogeneity would be accompanied by a substantial number of differentially expressed genes (DEGs). We compared the number of DEGs between Endo- and SV-enriched cells within different capillary sub-groups defined by either transcriptional states (i.e. clustering) or in different spatial locations, specifically, the proposed septal and non-septal cells of Cap1 and Cap2 (revised Figure 2g). If lineage was a major contributor to e17.5 heterogeneity, we would expect to see a substantial number of DEGs between Endo- and SV-enriched cells within a specific location. Instead, the results showed that the largest distinction is by far between Cap1 and Cap2 (202 DEGs) while comparing all Endo- vs. SV-derived cells identified only 24 DEGs. Importantly, once the effect of location was removed by only comparing Endo- and SV-enriched cells either inside or outside of the septum, there were only 6 DEGs between the lineages. Further supporting the impact of location on transcription, the second largest number of DEGs was between the septum and non-septum cells of Cap1, regardless of lineage. This is discussed in the text (page 29, lines 285-296).

Inspecting the DEG identities provided further support for the conclusion that Cap1 and Cap2 do not represent lineage-based heterogeneity. There are 202 DEGs between e17.5 Cap1 and Cap2 as compared to 24 DEGs between all e17.5 Endo- and SV-enriched capillaries. The latter genes are shown in revised Table 2.18 of these genes are also DEGs between Cap1 and Cap2. If the differential patterns of these genes were due to a lineage effect, we would expect them to have a greater log-fold change in the comparison between Endo and SV than in the comparison between Cap1 and Cap2. However, 16 of the 18 genes have a greater log fold change between Cap1 and Cap2 than between Endo- and SV-enriched (Table 2). This indicates that differential expression between the Endo- and SV-enriched capillaries mostly stems from the differential contribution of the Endo and SV lineages to Cap1 and Cap2. This observation was added to the text at pages 20-23, lines 221-233.

In total, we concluded that our analyses provided strong support that location is the predominant source of heterogeneity at e17.5. However, it does not ex­clude the possibility that there is a very small amount of lineage-based heterogeneity, born out in a small number of genes in a small number of cells that are not enough to contribute statistically significant effects on transcription. Therefore, we have changed the text throughout the paper to indicate this point and that we do not rule out that some minor degree of lineage-based heterogeneity exists at e17.5 (page 30, lines 309-311). We also discuss in the text that other experimental methods, such as ATACseq could reveal differences not detectable with scRNAseq (page 48, lines 517-518).

3. In Figure 1i-k, authors claim that Cap ECs of adult heart are relatively homogeneous. If so, how were they identified as distinct clusters? On the other hand, analysis of a previous study (Kalucka et al., Cell 2020) showed clear distinction between Cap1 and Cap2. Please describe the procedures for cluster identification in detail and explain the discrepancies between two datasets.

Since our adult dataset was processed using Smart-seq2 with greater sequencing depth but far fewer cells, which could have led to the discrepancies mentioned by the reviewer, we decided to sequence another *Bmx^CreER^*-lineage labeled adult dataset with many more cells using the 10X platform. Details are described below, but the main take away is that our conclusions remains the same—there is no transcriptional indication of lineage-based heterogeneity in adult ECs. To shorten the paper and avoid extraneous figures, we removed both the Smartseq2 data and our re-analysis of Kalucka et al., data (original Figure S5) from the re-submission, leaving only the new 10X data and a mention in the text regarding other datasets supporting our same findings on lineage (pages 12-15, lines 177-178).

The new 10x dataset contains ECs from *Bmx^CreER^* lineage-labeled adult mice (revised Figure 1i-k, Figure 1—figure supplement 1e, Figure 1—figure supplement 3c, Figure 1—figure supplement 5a, Figure 3h). Using the above-described criteria to set cluster resolution (major comment 1.1), this dataset revealed 3 clusters of capillary cells, one major cluster and two smaller clusters—one being marked by *Apln* and the other with interferon response genes, both of which were detected in Kalucka et al., (revised Figure 1i-k and Figure 1—figure supplement 5a). However, there was a similar distribution of cells into each of these clusters in both the Endo- and SV-enriched samples (Figure 1j-k), consistent with a convergence of EC lineages in the adult heart. This demonstrates that while some heterogeneity still exists among adult capillary cells, the cells from the Endo and SV lineages are transcriptionally equivalent. We have added these new results to the text at page 12, lines 168-177 and changed the wording throughout the text to indicate that while there is no heterogeneity between adult ECs from separate lineages, there is overall still heterogeneity among ECs in adult hearts.

4. The authors are required to describe in detail about the label transfer method and its limitations to follow up. It was employed for the comparison of human and mouse datasets, but this method could raise a biased characterization of the cell types. Therefore, the authors are strongly recommended to confirm the EC heterogeneity in human fetal hearts by IHC or ISH.

We appreciate the reviewers’ suggestion to validate the EC heterogeneity seen in the human fetal heart and investigate whether this heterogeneity is correlated with location, as it is in the mouse. To examine whether human Cap1 and Cap2 represent cells in different locations, we performed in situ hybridization for *TINAGL1*, a gene which is enriched in human Cap2 (revised Figure 5i). This revealed a statistically significant difference in the amount of *TINAGL1* RNA detection between the septum and the heart walls in both an 18 week and a 20 week gestational heart, with the septum having dramatically lower expression (revised Figure 5j-k). This difference was especially pronounced between the septum and the right ventricular free wall, similar to the Car4 pattern in mouse (Figure 3e and Figure 3—figure supplement 1b). Thus, in situ hybridization with *TINAGL1* supports a bias in the localization of human Cap1 to the septum and human Cap2 to the ventricular wall. This data in combination with the shared transcriptional patterns between mouse and human, especially in the expression of hypoxia-induced genes (Figure 5f-h and Figure 1—figure supplement 1d), allowed us to conclude that the location-based transcriptional effects present during mouse development are likely also present during human development. These results were added to the text at page 40, lines 388-398.

In regards to describing the label transfer method, it is described in detail by Stuart et al., (Stuart et al., 2019). Briefly, diagonalized canonical correlation analysis is used for dimensionality reduction of both datasets, followed by L2 normalization. Then, a mutual nearest neighbors approach is used to identify pairs of cells (“anchors”) between the two datasets that represent a similar biological state. Every cell in the query dataset is assigned an anchor in the reference dataset (with an associated anchor score), and the cluster label of the reference cell is assigned to the query cell.

The major limitation of this method as we applied it is that it compares datasets from different species that were generated using different methods (Smart-seq2 and 10x) and different sequencing depths. These differences, as well as limitations associated with these methods, impact accuracy. For example, most of the mouse vein cells match to human Cap2 rather than human veins (Figure 5g), likely because there are so few human vein cells while cells in the larger Cap2 population have a larger local neighborhood to strengthen their anchor score. In addition, the matching is limited to the clusters that are pre-annotated in each dataset (i.e., each human cell will match to its closest mouse cluster, even if there is not a true biological correlate). Finally, genes used in the mapping were limited to those that had a 1:1 homology mapping between mouse and human, meaning that some information was eliminated before label transfer. Despite these limitations, we provided evidence strengthening the correspondence between the human and mouse capillary clusters by: 1. the observation of shared patterns of gene expression (Figure 5h and Figure 5—figure supplement 1d) and 2. In situ validation of select genes (Figure 3b-g and 5i-k). Description of the method and its limitations are discussed in the *Methods* at pages 61-62, lines 732-750.

5. The inference of developmental pathway using the trajectory analysis is not straightforward. In Figure 6b, the trajectory analysis predicts development from Art3 to Art2 and Art1, which does not match with their localizations as shown in Figure 7f-g. Authors need to describe how they made the direction of the flow in the trajectory analysis.

The direction of the arterial trajectory was determined by the arrows from the RNA velocity analysis, which provides this directional information as a consequence of comparing spliced (mature) to unspliced (immature) transcripts (La Manno et al., 2018). Additionally, this trajectory is supported by previous lineage analyses in mouse of a trajectory from capillaries to *Gja5*- arteries to *Gja5*+ arteries (Su et al., 2018). This trajectory is also consistent with the localization showed in Figure 7f-g, which shows that *GJA5*+ arteries in human are larger and more proximal than *GJA5*- arteries, as they are in mice. Finally, the interpretation of Art3 being a less mature artery population is supported by its disappearance in the adult human (Figure 6c). Two explanations for this are that: (1) less mature artery cells coalesce or migrate into larger arteries and take on a more mature phenotype, or (2) vessels composed of less mature artery cells stay the same size but eventually get exposed to more flow or some other factors which induces maturation. A more clear explanation is now included at page 41, lines 426-430 and page 44, lines 479-481.

Reviewer #3 (Recommendations for the authors):The authors claim is well supported by their data. scRNAseq and histological examination clearly support their idea "convergent differentiation" of coronary ECs, although the results shown in the present study are descriptive. There are several points that should be addressed before publication.1. According to the previous reports by several groups including the authors' group, the SV-derived cells and the Endo-derived cells contribute to the vessels in the outer myocardial wall and inner myocardial wall. Now it is striking difference of the gene expression of ECs between ventral and dorsal wall of myocardium (Figure 3). Neither blood flow nor ischemia might not account for this regional difference at e17.5. If possible, the authors are encouraged to explain or discuss the difference of dorsal and ventral wall, although both are outer layer of myocardium.

Previous reports by our lab and Dr. Bin Zhou at the Shanghai Institute of Biochemistry and Cell Biology reveal a slightly more nuanced picture that just outer and inner myocardial wall distinction. First, there is an invasion of ECs into the dorsal wall derived from the SV. Then, the Endo invades the septum to produce vessels that subsequently migrate heavily towards the ventral wall. Thus, during development, there is a large lineage distinction between the dorsal and ventral sides (even though there is some mixing, Tian et al., 2015). After birth, these two sources expand in a way that leads to a final contribution where SV-derived vessels are skewed to the outer wall while Endo-derived vessels are skewed to the inner wall around the entire heart, although there are some differences in the extent of skewing at different locations.

We hypothesize that blood flow could account for differences between the dorsal and ventral walls because of the timing and degree of vascularization that occurs during development, particularly since the precise timing of Endo-derived vessels integrating into the patent coronary circulation has not been formally demonstrated. As described above, we previously showed that SV-derived coronary vessels are more numerous and provide more coverage on the dorsal side of the heart compared to the ventral side at least until e15.5 (Red-Horse et al., 2010; Chen et al., 2014). Therefore, whatever environmental variables related to blood supply (including flow and hypoxia) distinguish the septum and the dorsal wall at e17.5 likely also cause the differences between the ventral and dorsal walls show in Figure 3d-g. At this stage in development, the anatomic similarity of both dorsal and ventral being outer myocardium during development may be less significant physiologically than the timing and degree of their vascularization. This interpretation was added to the text at page 47, lines 494-501.

2. To compare the gene expression of ECs between mouse and human, the authors used the Seurat label Transfer workflow to reference map each human cell to its closest mouse cluster (Figure 5e-g). Indeed, human Cap1 might be a cluster enriched of septal cells with less blood flow. Therefore, the authors claim that similar environment (ischemia and flow) between mouse and human contributes to clustering of coronary ECs.Given the methods described for human heart sections (line 707-711), human tissues are available for examining expression of genes listed in Figure S10c or 10d as that of mice (Kcne3) was examined in Figure 3f. Histological examination of gene or protein expression using human hearts to investigate where genes liste in Figure S10 would support the authors' claim.

We appreciate the reviewers’ suggestion to validate the EC heterogeneity seen in the human fetal heart and investigate whether this heterogeneity is correlated with location, as it is in the mouse. To examine whether human Cap1 and Cap2 represent cells in different locations, we performed in situ hybridization for *TINAGL1*, a gene which is enriched in human Cap2 (revised Figure 5i). This revealed a statistically significant difference in the amount of *TINAGL1* RNA detection between the septum and the heart walls in both an 18 week and a 20 week gestational heart, with the septum having dramatically lower expression (revised Figure 5j-k). This difference was especially pronounced between the septum and the right ventricular free wall, similar to the Car4 pattern in mouse (Figure 3e and Figure 3—figure supplement 1b). Thus, in situ hybridization with *TINAGL1* supports a bias in the localization of human Cap1 to the septum and human Cap2 to the ventricular wall. This data in combination with the shared transcriptional patterns between mouse and human, especially in the expression of hypoxia-induced genes (Figure 5f-h and Figure 5—figure supplement 1d), allowed us to conclude that the location-based transcriptional effects present during mouse development are likely also present during human development. These results were added to the text at page 40, lines 388-398.

3. Related to 1 and 2, the distinct gene expression between inner and outer myocardial wall and between dorsal and ventral wall found in mouse hearts could be examined by immunohistochemical or in situ hybridization analyses using human embryonic hearts. These experiments strengthen the conclusion of comparable "convergent differentiation of coronary Cap ECs" between mouse and human.

As described above, we observed the expression of *TINAGL1*, a marker gene distinguishing human capillary clusters, in different regions of the human fetal heart. In contrast to mouse Car4, there was no difference in *TINAGL1* detection between the ventral and dorsal walls at 18 and 20 weeks (Figure 5k). One possible explanation for this is that the ventral and dorsal walls in human have similar environmental conditions at this stage, and thus a similar distribution of cells in the Cap1 and Cap2 state. Since the dorsal-ventral differences in mouse are likely due to the mismatched timelines of angiogenesis from the, SV and Endo, the lack of an apparent difference between ventral and dorsal gene expression patterns in human could indicate that the timing of vascularization is more consistent throughout the developing human heart. This interpretation was added to the text at page 48, lines 530-533.